# Learning Realistic Traffic Agents in Closed-loop

**Chris Zhang**   **James Tu**   **Lunjun Zhang**   **Kelvin Wong**   **Simon Suo**[*]   **Raquel Urtasun**

Waabi     University of Toronto
{czhang,jtu,lzhang,kwong,urtasun}@waabi.ai

**Abstract:** Realistic traffic simulation is crucial for developing self-driving software in a safe and scalable manner prior to real-world deployment. Typically, imitation learning (IL) is used to learn human-like traffic agents directly from real-world observations collected offline, but without explicit specification of traffic rules, agents trained from IL alone frequently display unrealistic infractions like collisions and driving off the road. This problem is exacerbated in out-of-distribution and long-tail scenarios. On the other hand, reinforcement learning (RL) can train traffic agents to avoid infractions, but using RL alone results in unhuman-like driving behaviors. We propose Reinforcing Traffic Rules (RTR), a holistic closed-loop learning objective to match expert demonstrations under a traffic compliance constraint, which naturally gives rise to a joint IL + RL approach, obtaining the best of both worlds. Our method learns in closed-loop simulations of both nominal scenarios from real-world datasets as well as procedurally generated long-tail scenarios. Our experiments show that RTR learns more realistic and generalizable traffic simulation policies, achieving significantly better tradeoffs between human-like driving and traffic compliance in both nominal and long-tail scenarios. Moreover, when used as a data generation tool for training prediction models, our learned traffic policy leads to considerably improved downstream prediction metrics compared to baseline traffic agents.

**Keywords:** Traffic simulation, Imitation learning, Reinforcement learning

## 1   Introduction

Simulation is a critical component to safely developing autonomous vehicles. Designing realistic traffic agents is fundamental in building high-fidelity simulation systems that have a low domain gap to the real world. However, this can be challenging as we need to both capture the idiosyncratic nature of *human-like* driving and avoid unrealistic traffic *infractions* like collisions or driving off-road. Existing approaches used in the self-driving industry lack realism: they either replay logged trajectories in a non-reactive manner [1, 2] or use heuristic policies which yield rigid, unhuman-like behaviors. Using data-driven approaches to learn more realistic policies is a promising alternative.

The dominant data-driven approach has been imitation learning (IL), where nominal human driving data is used as expert supervision to train the agents. However, while expert demonstrations provide supervision for human-like driving, pure IL methods lack explicit knowledge of traffic rules and infractions which can result in unrealistic policies. Furthermore, the reliance on expert demonstrations can be a disadvantage, as long-tail scenarios with rich interactions are very rare, and thus learning is overwhelmingly dominated by more common scenarios with a much weaker learning signal.

Reinforcement learning (RL) approaches encode explicit knowledge of traffic rules through hand-designed rewards that penalize infractions [3, 4, 5, 6, 7]. These approaches do not rely on expert demonstrations and instead learn to maximize traffic-compliance rewards through trial and error. In the context of autonomy, this allows training on synthetic scenarios that do not have expert demonstrations in order to improve the robustness of learned policies [5]. However, traffic rules alone cannot describe all the nuances of human-like driving, and it is still an open question if one can manually design a reward that can completely capture those intricacies.

---

[*]Work done at Waabi.

7th Conference on Robot Learning (CoRL 2023), Atlanta, USA.

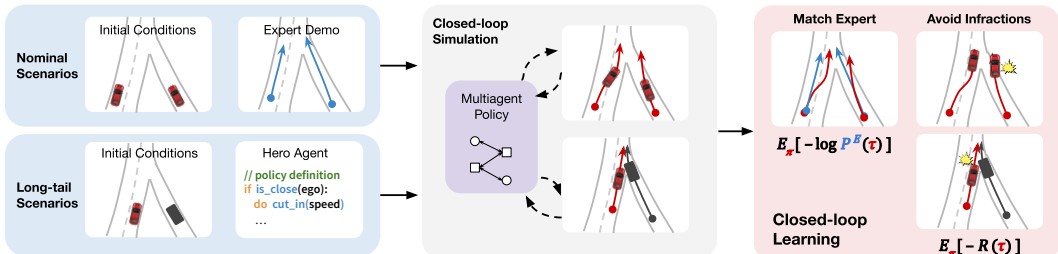

Figure 1: Our multi-agent policy is trained in closed-loop to match expert demonstrations under a traffic compliance constraint using both nominal offline data and simulated long-tail scenarios as a rich learning environment. This gives rise to an IL objective which supervises the policy using real-world expert demonstrations and an RL objective which explicitly penalizes infractions.

Towards learning human-like and traffic-compliant agents, we propose Reinforcing Traffic Rules (RTR), a holistic closed-loop learning method to match expert demonstrations under a traffic-compliance constraint using both nominal offline data and additional simulated long-tail scenarios (Figure 1). We show our formulation naturally gives rise to a unified closed-loop IL + RL objective, which we efficiently optimize by exploiting differentiable dynamics and a per-agent factorization. In contrast to prior works that combine IL and RL [1], our closed-loop approach allows the model to understand the effects of its actions and suffers significantly less from compounding error. Furthermore, exploiting simulated long-tail scenarios improves learning by exposing the policy to more interesting interactions that would be difficult and possibly dangerous to collect from the real world at scale. Our experiments show that unlike a wide range of baselines, RTR learns realistic policies that better generalize to both nominal and long-tail scenarios *unseen during training*. The benefits carry forward to *downstream tasks* such as simulating scenarios to train autonomy models; prediction models trained on data simulated with RTR have the strongest prediction metrics on real data, serving as further evidence that RTR has learned more realistic traffic simulation. We believe this serves as a crucial step towards more effective applications of traffic simulation for self-driving.

## 2 Related Work

**Traditional traffic simulation:** To generate general traffic flow, simulators [8, 9, 10, 11] typically use heuristic models [12, 13, 14] as models of human driving. While these heuristic models are useful in capturing high-level traffic characteristics like flow and density, they are lacking in capturing the lower-level nuances of human driving, thus limiting their applicability in self-driving. For more realistic traffic models, we explore using machine learning as a more promising approach.

**Imitation learning:** IL methods learn a control policy from expert demonstrations. In the context of autonomous vehicles, [15] pioneered the use of behavior cloning (BC) to learn a driving policy in open-loop. Since then, open-loop methods have been explored for both autonomy [16, 17, 18] and traffic simulation [19, 20, 21]. Open-loop methods primarily suffer from distribution shift due to compounding error [22], and so various techniques like data augmentation [23, 16], uncertainty-based regularization [24, 25], and augmentation with a rules-based planner [21] have been proposed to alleviate the problem. Closed-loop imitation learning approaches [26, 27, 28, 29], which address distribution shift by exposing the policy to a self-induced state distribution during training, have also been explored in traffic simulation [30, 31, 32, 33]. While IL exploits expert demonstrations, there is a lack of explicit knowledge on safety-critical aspects like avoiding infractions. Methods like differentiable common-sense penalties [30, 23], additional finetuning [34], and test-time guided sampling [35] have been proposed to complement the standard IL approach. In this work, we use reinforcement learning to explicitly encode general non-differentiable traffic rules.

**Reinforcement learning:** RL methods [36, 37, 38] do not require expert demonstrations and instead learn through interacting with the environment and a reward function. In self-driving, knowledge of infractions can be encoded in the reward [3, 4, 5, 6, 7]. Because RL does not require expert demonstrations, it is possible to train on procedurally generated scenarios for improved infraction avoidance [5]. However, it is difficult to learn realistic driving behavior using reward alone. RL

methods can be sample inefficient [3, 4], and specifying human-like driving with a scalar reward is difficult. In this work, we supplement RL with IL to learn more human-like driving while still enjoying the explicit learning signal provided from the reward.

**Combined IL + RL:** Pretrained IL policies can be used as initialization to guide exploration [39, 40] or regularize learning [41, 42, 43, 44], and offline data can be used to bootstrap learning and help with sparse rewards [45, 46]. Offline RL methods also use IL for out-of-distribution generalization and overestimation [47, 48]. In self-driving, IL has been used as a pre-training phase improve sample efficiency [49]. Recent work augments open-loop IL with RL [1, 50, 51] to learn more robust models. While promising, the open-loop nature of BC leaves the policy susceptible to distribution shift. In this work, we explore a holistic closed-loop IL + RL method for traffic simulation.

**Long-tail Scenarios:** Real data can be curated [1, 52, 53] for more interesting scenarios, but collecting these at scale can be unsafe and expensive. Alternatively, scenarios can be generated by maximizing an adversarial objective w.r.t. to the ego [54, 55, 56], but incorporating factors like diversity for training scenarios is still an open problem. In this work, we use knowledge-based approaches [57, 58] to guide generation towards a large variety of difficult but realistic scenarios.

## 3 Learning Infraction-free Human-like Traffic Agents

To learn realistic infraction-free agents, we propose a unified learning objective to match expert demonstrations under an infraction-based constraint. We show how our formulation naturally gives rise to a joint closed-loop IL + RL approach which allows learning from both offline collected human driving data when possible, and additional simulated long-tail scenarios containing rich interactions that would otherwise be difficult or impossible to collect in the real world.

### 3.1 Preliminaries

We model multi-agent traffic simulation as a Markov Decision Process $\mathcal{M} = (\mathcal{S}, \mathcal{A}, R, P, \gamma)$ with state space, action space, reward function, transition dynamics, and discount factor respectively. As our focus is *traffic simulation* where we have access to all ground truth states, we opt for a fully observable and centralized multi-agent formulation where a single model jointly controls all agents. This enables efficient inference by sharing computation [2], and easier interaction modeling.

**State, action and policy:** We define the state $s = \{s^{(1)}, \ldots, s^{(N)}, m\} \in \mathcal{S}$ to be the joint states of $N$ agents where $N$ may vary across different scenarios, as well as an HD map $m$ which captures the road and lane topology. We parameterize the state of the $i$-th agent $s^{(i)}$ with its position, heading, and velocity over the past $H$ history timesteps. The state also captures 2D bounding boxes for each agent. Likewise, $a = \{a^{(1)}, \ldots, a^{(N)}\} \in \mathcal{A}$ is the joint action which contains the actions taken by all the agents. The $i$-th agent's action $a^{(i)}$ is parameterized by its acceleration and steering angle. Agents are controlled by a single centralized policy $\pi(a|s)$ which maps the joint state to joint action.

**Trajectories and dynamics:** We define a trajectory $\tau_{0:T} = (s_0, a_0, \ldots, s_{T-1}, a_{T-1}, s_T)$ as a sequence of state action transitions of length $T$ for all agents. We use the kinematic bicycle model [59] as a simple but realistic model of transition dynamics $P(s_{t+1}|s_t, a_t)$ for each agent. Trajectories can be sampled by first sampling from some initial state distribution $\rho_0$ before unrolling a policy $\pi$ through the transition dynamics, i.e. $P^\pi(\tau) = \rho_0(s_0) \prod_{t=0}^{T-1} \pi(a_t|s_t) P(s_{t+1}|s_t, a_t)$.

**Reward:** Let $R^{(i)}(s, a^{(i)})$ be a per-agent reward which is specific for the $i$-th agent, but dependent on the state of all agents, to model interactions such as collision. The joint reward is then $R(s, a) = \sum_i^N R^{(i)}(s, a^{(i)})$, with $R(\tau) = \sum_{t=0}^{T-1} \gamma^t R(s_t, a_t)$ as the $\gamma$-discounted return of a trajectory.

**Policy learning:** Both imitation learning (IL) and reinforcement learning (RL) can be described in this framework. IL can be described as an $f$-divergence minimization problem: $\pi^* = \arg\min_\pi D_f\left(P^\pi(\tau) \parallel P^E(\tau)\right)$ where $P^E$ is the expert-induced distribution. RL on the other hand aims to find the policy which maximizes the expected reward $\pi^* = \arg\max_\pi \mathbb{E}_{P^\pi}[R(\tau)]$.

---

[2]In our experiments, our model easily scales to 50 agents and a map ROI of $1000m \times 400m$ per simulation.

## 3.2 Learning

To learn a multiagent traffic policy that is as human-like as possible while avoiding infractions, we consider the reverse KL divergence to the expert distribution with an infraction-based constraint

$$\underset{\pi}{\arg\min} \quad D_{\mathrm{KL}}\left(P^{\pi}(\tau) \parallel P^{E}(\tau)\right) \qquad (1) \qquad R^{(i)}(\boldsymbol{s}, a^{(i)}) = \begin{cases} -1 & \text{if infraction} \\ 0 & \text{otherwise,} \end{cases} \qquad (2)$$
$$\text{s.t.} \quad \mathbb{E}_{P^{\pi}}\left[R(\tau)\right] \geq 0$$

where $R^{(i)}$ is a per-agent reward function that penalizes any infractions (collision and off-road events). For a rich learning environment, we consider both a dataset $D$ of nominal expert trajectories $\tau^{E} \sim P^{E}$ collected by driving in the real world, and additional simulated *long-tail scenarios*. Unlike real world logs, these scenarios contain what we denote as *hero* agents, which induce interesting interactions like sudden cut-ins, etc. (details in Section 3.4). More precisely, let $\pi_{\theta}$ be our learner policy. Let $\boldsymbol{s}_0^S \sim \rho_0^S$ be the initial state sampled from the long-tail distribution and $\pi_{\boldsymbol{s}_0}^S$ represent the policy of the hero agent. The overall multiagent policy is given as

$$\pi(\boldsymbol{a}_{i,t}|\boldsymbol{s}_t) = \begin{cases} \pi_{\boldsymbol{s}_0}^S(a_t^{(i)}|\boldsymbol{s}_t) & \text{if agent } i \text{ is hero} \\ \pi_{\theta}(a_t^{(i)}|\boldsymbol{s}_t) & \text{otherwise.} \end{cases} \qquad (3)$$

The overall initial state distribution is then given as $\rho_0 = (1-\alpha)\rho_0^D + \alpha\rho_0^S$, where $\rho_0^D$ corresponds to the offline nominal distribution, and $\alpha \in [0, 1]$ is a hyperparameter that balances the mixture.

Taking the Lagrangian of Equation 8 decomposes the objective into an IL and RL component,

$$\mathcal{L} = \mathbb{E}_{P^{\pi}}\left[\underbrace{-\log P^{E}(\tau)}_{\text{IL}} - \lambda \underbrace{R(\tau)}_{\text{RL}}\right] - H(\pi) = \mathcal{L}^{\mathrm{IL}} + \lambda\mathcal{L}^{\mathrm{RL}} - H(\pi) \qquad (4)$$

where $\lambda$ is a hyperparameter balancing the two terms, and $H(\pi)$ is an additional entropy regularization term [3] [26]. Notably, we optimize IL and RL *jointly in a closed-loop manner*, as the expectation is taken with respect to the on-policy distribution $P^{\pi}(\tau)$. Compared to open-loop behavior cloning, the closed-loop IL component allows the model to experience states induced by its own policy rather than only the expert distribution, increasing its robustness to distribution shift. Furthermore, while the additional reward constraint may not change the optimal solution of the unconstrained problem (the expert distribution may be infraction-free), it can provide additional learning signal through RL.

The RL component $\mathbb{E}_{P^{\pi}}\left[R(\tau)\right]$ can be optimized using standard RL techniques and exploits both offline-collected nominal scenarios and simulated long-tail scenarios containing rich interactions. However, the imitation component $\mathcal{L}^{\mathrm{IL}}$ is only well-defined when expert demonstrations are available and thus only applied to nominal data. We start from an initial state $\boldsymbol{s}_0^E \sim \rho_0^D$ and have the policy $\pi_{\theta}$ control all agents in closed-loop simulation. The loss is the distance between the ground truth and policy-induced trajectory [4].

$$\mathcal{L}^{\mathrm{IL}} = \mathbb{E}_{\tau^E \sim D}\left[\mathbb{E}_{\tau \sim P^{\pi}(\cdot|\boldsymbol{s}_0^E)}\left[D(\tau^E, \tau)\right]\right]. \qquad (5)$$

It is difficult to obtain accurate action labels for human driving data in practice, so we only consider states in our loss, i.e. $D(\tau^E, \tau) = \sum_{t=1}^{T} d(\boldsymbol{s}_t^E, \boldsymbol{s}_t)$ where $d$ is a distance function (e.g. Huber).

**Optimization:** To optimize Equation 4, we first note that the $\mathcal{L}^{IL}$ component is differentiable by using the reparameterization trick [60] when sampling from the policy[5] and differentiating through the transition dynamics (kinematic bicycle model). We refer the reader to the appendix for more details. To optimize the $\mathcal{L}^{RL}$ component, we design a centralized and fully observable variant of PPO [36]. While it is possible to directly optimize the policy with the overall scene reward

---

[3] The causal entropy term is included as an entropy regularizer in some learning algorithms such as PPO [36]. In our setting, we empirically found that it was not necessary to include.

[4] As we do not have access to $P^E$ directly to query log-likelihood, using a distance is essentially making the assumption that $P^E(\tau) \propto \exp\left[-D(\tau^E, \tau)\right]$.

[5] We found that directly using the mean action provides good results without the need for sampling.

$R(\boldsymbol{s}, \boldsymbol{a}) = \sum_{i=1}^{N} R^{(i)}(\boldsymbol{s}, a^{(i)})$, we instead optimize each agent individually with their respective individual reward $R_i(\boldsymbol{s}, a_i)$. While this factorized approach may ignore second-order interaction effects, it considerably simplifies the credit assignment problem leading to more efficient learning. More precisely, we compute factorized value targets $V^{(i)} = \sum_{t=0}^{T} \gamma^t R_t^{(i)}(\boldsymbol{s}_t, a_t^{(i)})$, and the factorized PPO policy loss is given as $\mathcal{L}^{\text{policy}} = \sum_{i=1}^{N} \min(r^{(i)} A^{(i)}, \text{clip}(r^{(i)}, 1 - \epsilon, 1 + \epsilon) A^{(i)})$ where the probability ratio is factorized, i.e. $r^{(i)} = \pi(a^{(i)}|\boldsymbol{s}, \boldsymbol{m})/\pi_{\text{old}}(a^{(i)}|\boldsymbol{s}, \boldsymbol{m})$ and $A^{(i)}$ is a factorized GAE [61] estimate. More details can be found in the appendix.

### 3.3 Model Architecture

Our traffic model $\pi_\theta$ architecture uses common ideas from SOTA traffic agent motion forecasting literature in order to extract context and map features and predict agent actions (Figure 2). Recall that a state $\boldsymbol{s} = \{s^{(1)}, \ldots, s^{(N)}, \boldsymbol{m}\}$ consists of each individual agent's states $s^{(i)}$ that contain the agent's kinematic state over a history horizon $H$, and an HD map $\boldsymbol{m}$. From each agent's state history, a shared 1D CNN and GRU are used to extract agent history context features $h_a^{(i)} = f(s^{(i)})$. At the same time, a GNN is used to extract map features from a lane graph representation of the

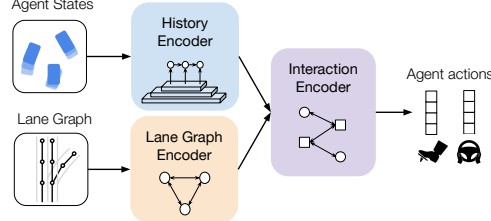

Figure 2: Our multiagent policy architecture. The value network architecture is the same but regresses value targets instead.

map input $h_m = g(\boldsymbol{m})$. A HeteroGNN [62] then jointly fuses all agent context features and map features before a shared MLP decodes actions for each agent independently.

$$\{h^{(1)}, \ldots h^{(N)}\} = \text{HeteroGNN}(\{h_a^{(1)}, \ldots, h_a^{(N)}\}, h_m) \tag{6}$$

$$(\mu^{(i)}, \sigma^{(i)}) = \text{MLP}(h^{(i)}). \tag{7}$$

We use independent normal distributions to represent the joint agent policy, i.e. $\pi(a^{(i)}|\boldsymbol{s}) = \mathcal{N}(\mu^{(i)}, \sigma^{(i)})$, and thus $\pi(\boldsymbol{a}|\boldsymbol{s}) = \prod_{i=1}^{N} \pi(a^{(i)}|\boldsymbol{s})$. Note that agents are only independent *conditional* on their shared past context, and thus important interactive reasoning is still captured. Our value model uses the same architecture but does not share parameters with the policy; we compute $\{h_0^v, \ldots, h_N^v\}$ in a similar fashion, and decode per-agent value estimates $\hat{V}^{(i)} = \text{MLP}^v(h^{(i)})$.

### 3.4 Simulated Long-tail Scenarios

Nominal driving logs can be monotonous and provide weak learning signal when repeatedly used for training. In reality, most traffic infractions can be attributed to rare and long-tail scenarios belonging to a handful of scenario families [63] which can be difficult and dangerous to collect from the real world at scale. In this work, we procedurally generate long-tail scenarios to supplement nominal logs for training and testing. Following the self-driving industry standard, we use *logical scenarios* [64, 57] which vary in the behavioral patterns of particular hero agents with respect to an ego agent (e.g. cut-in, hard-braking, merging, etc.). Designed by expert safety engineers, each logical scenario is parameterized by $\theta \in \Theta$ which controls lower-level aspects of the scenario such as behavioral characteristics of the hero agent (e.g. time-to-collision or distance triggers, aggressiveness, etc.), exact initial placement and kinematic states, and geolocation. A *concrete scenario* can then be procedurally generated in an automated fashion by sampling a logical scenario and corresponding parameters $\theta$. While these scenarios cannot be used for imitation as they are simulated and do not have associated human demonstrations, they provide a rich reinforcement learning signal due to the interesting and rare interactions induced by the hero agents.

## 4 Experiments

**Scenario sets:** Our experiments use two datasets that represent nominal and long-tail scenarios respectively. The NOMINAL dataset consists of a set of highway logs which capture varying traffic densities and road topologies while containing expert demonstrations. The dataset consists of 465

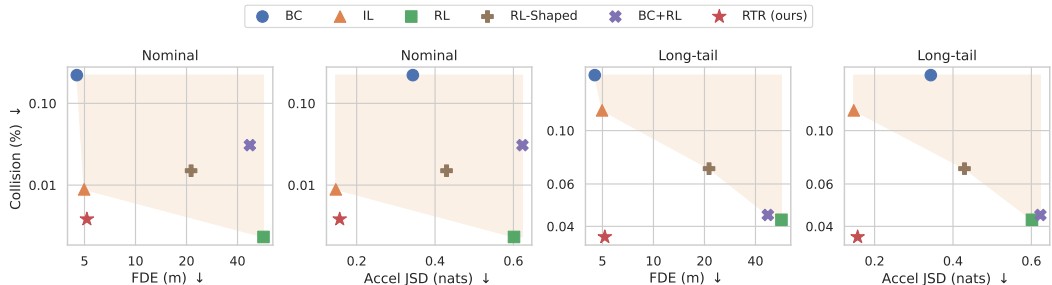

Figure 3: Metrics (lower is better) on held-out nominal and long-tail scenarios. Pareto frontier of baselines is shaded; RTR achieves the best tradeoff between infraction and other realism metrics.

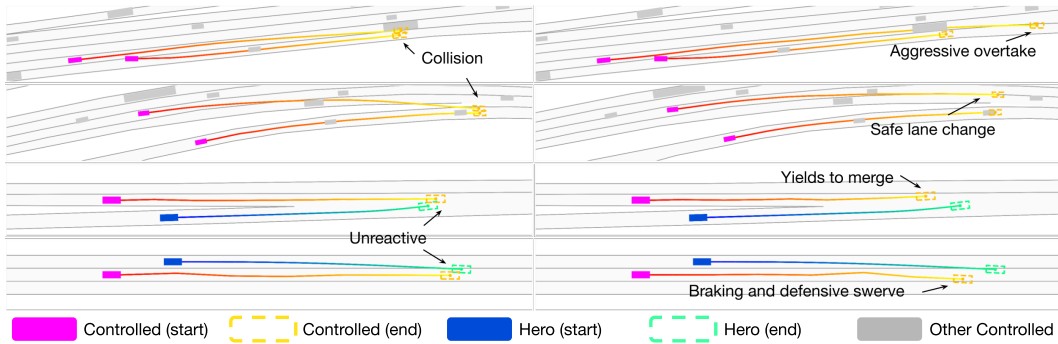

Figure 4: Qualitative examples comparing the baseline IL model (left) and RTR (right). Scenarios with hero agents (blue) are from the long-tail set. All other agents are controlled; pink is used for visual emphasis. RTR avoids infractions while maintaining diverse, human-like driving behavior.

snippets for training and 115 for testing, where each snippet lasts for 20 seconds. We use LONGTAIL to denote the scenario set generated using the process outlined in Section 3.4 which contain rare actor maneuvers like sudden cut-ins. We use 25 logical scenarios to generate a total of 333 concrete scenarios, where 167 concrete scenarios are used for training and 166 are held-out for evaluation. This evaluation set is held-out on the parameter level and measures in-distribution generalization. We also evaluate on an additional set of held out logical scenarios to measure out-of-distribution generalization, with more details in Section 4.1.

**Metrics:** We evaluate our traffic models' ability to 1) match human-like driving and 2) avoid infractions. For the former, we measure similarity to the demonstration data by computing the final displacement error (**FDE**) [33, 30], which measures the L2 distance between the agent's simulated and ground truth (GT) position after 5 seconds. Furthermore, we use Jensen-Shannon Divergence (**JSD**) [33, 31] between histograms of scenario features (agent acceleration) in order to measure distributional realism. Finally, to measure infraction rates, we consider **collision** and driving **off-road**. We use a bootstrap resampling over evaluation snippets to compute uncertainty estimates. Results with more extensive metrics (and their definitions) can be found in the appendix.

## 4.1 Benchmarking Traffic Models

**Comparison to state-of-the-art:** We evaluate RTR and several baselines on both nominal and long-tail scenarios. For comparability, we use the same input representation and model architecture as described in Section 3.3 for all methods. Our first two baselines are representative of state-of-the-art imitation learning approaches for traffic simulation. **BC** is our single-step behavior cloning baseline following [19]. The **IL** baseline is trained using closed-loop policy unrolling [30, 31]. Next, **RL** is trained using our proposed factorized version of PPO [36] with the reward in Equation 20. The **RL-Shaped** baseline includes an additional reward for driving at the speed-limit to encourage more human-like driving. Finally, **BC+RL** is an RL augmented BC baseline following [1].

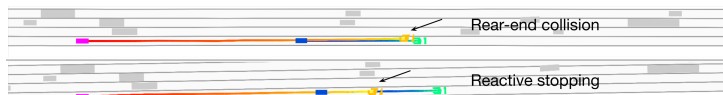

| Meth. | Col. (%) | Off. (%) |
|-------|----------|----------|
| IL    | $11.8 \pm 2.1$ | $1.0 \pm 0.1$ |
| RTR   | $\mathbf{5.0 \pm 1.4}$ | $\mathbf{0.3 \pm 0.1}$ |

Figure 6: IL (top), RTR (bottom) on an out-of-distribution scenario where a hero agent (blue) comes to a complete stop on the highway.

Table 1: Results on out-of-distribution long-tail set.

Figures 3 and 4 show the results; a full table can be found in the appendix. Firstly, the BC model achieves poor realism because it suffers from distribution shift during closed-loop evaluation as it encounters states unseen during training due to compounding error. Next, we see RL achieves low infraction rates but results in unhuman-like driving (Figure 5). This is because it is difficult for reward alone to capture realistic driving. Efforts in reward shaping result in improvements but are ultimately still insufficient. We see BC+RL improves upon BC

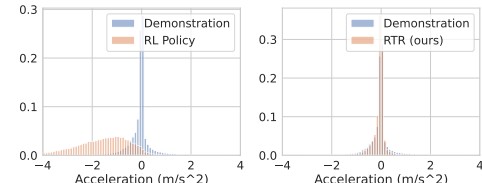

Figure 5: RL policy naively decelerates to avoid infractions. RTR learns to avoid collision more naturally without slowing down.

infractions but still lacks realism. This is because BC is an open-loop objective and only provides signal in expert states, while only the RL signal is present in non-expert states. Thus, the policy still suffers from compounding error with respect to imitation. On the other hand, closed-loop IL performs better as it is more robust to compounding error, but still struggles on the long-tail scenario set without explicit supervision. Finally, the *holistic* closed-loop IL and RL approach of RTR improves infraction rates while maintaining reconstruction and JSD metrics. We see RTR outperforms even pure RL in terms of infraction rate on long-tail scenarios, suggesting that including long-tail scenarios during training can help the model generalize to held-out evaluation long-tail scenarios.

**Out-of-distribution generalization:** Recall from Section 3.4 that logical scenarios define a family of scenarios and concrete scenarios define variations within a family. While we have evaluated in-distribution generalization by using held-out concrete scenarios, we further evaluate on *held-out logical scenarios*. We use 11 held-out logical scenarios with new map topologies and behavioral patterns to procedurally generate an additional out-of-distribution set consisting of 84 concrete scenarios. Our results show that RTR generalizes to this set better than baselines (Figure 6, Table 1).

### 4.2 Downstream Evaluation

One downstream application of traffic simulation is generating synthetic data for training autonomy models. We evaluate if the improved realism of RTR transfers in this context. Each model is used to generate a synthetic dataset of 589 scenarios which we use to train a SOTA prediction model [62] before evaluating its performance on held-out real data. Besides FDE, the cross-track error (CTE) of predicted trajectories projected onto the GT are used as prediction metrics. More experiment details can be found in the appendix. Table 2 shows that using RTR to generate training data results in the best prediction model. This provides evidence that RTR has learned more realistic behavior and has a lower domain gap compared to baselines, showing that our approach can

| Method | FDE (m) | CTE (m) |
|--------|---------|---------|
| BC     | $2.44 \pm 0.05$ | $0.90 \pm 0.04$ |
| IL     | $1.75 \pm 0.06$ | $0.28 \pm 0.01$ |
| RL     | $15.42 \pm 1.21$ | $0.32 \pm 0.02$ |
| RL-Shp | $6.66 \pm 0.26$ | $0.33 \pm 0.01$ |
| BC+RL  | $9.06 \pm 0.50$ | $0.42 \pm 0.03$ |
| RTR    | $\mathbf{1.58 \pm 0.05}$ | $\mathbf{0.27 \pm 0.03}$ |

Table 2: Prediction model trained on synthetic, evaluated on real.

improve the application of traffic simulation in developing autonomous vehicles.

### 4.3 Additional Analysis

**Long-tail scenarios:** We evaluate our approach of using procedurally generated scenarios against the alternative of mining hard scenarios from data [1, 53] by curating a set of logs from NOMINAL that the IL model commits an infraction on. Figure 7 shows that using only curated scenarios does not transfer well to the long-tail set, and in fact introduces a regression in the nominal scenarios, sug-

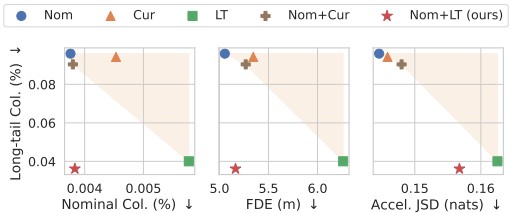

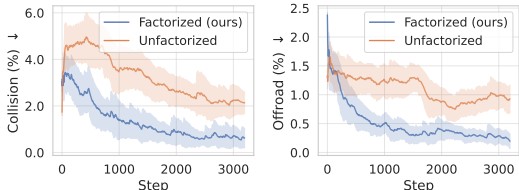

Figure 7: Using both nominal and long-tail yields the best tradeoff compared to baselines.

Figure 8: Our factorized PPO vs. standard PPO which uses a single scene-level reward.

gesting the model is overfitting to the curated scenarios. Up-sampling curated scenarios (Nom+Cur) also fails – relying purely on offline data may require prohibitively larger scale data collection.

**Factorized multiagent RL:** To ablate our factorized per-agent approach to multiagent PPO, we compare to a standard PPO implementation where the scene-level reward $R(\boldsymbol{s}, \boldsymbol{a})$ is used as supervision for the joint policy rather than each individual agent reward $R^{(i)}(\boldsymbol{s}, a^{(i)})$. Figure 8 shows that the factorized loss outperforms the alternative, likely due to the fact that multiagent credit assignment is extremely difficult when using the scene-level reward, leading to poor sample efficiency.

**Balancing the trade-off:** Recall from Section 3.2 that RTR balances human-like driving and avoiding infractions by weighting the IL vs. RL loss with $\lambda$ and nominal vs. long-tail training with $\alpha$ (e.g. $\lambda = \alpha = 0$ is the IL baseline). We found increasing the relative weight of RL and long-tail scenarios generally improves infraction avoidance while increasing the relative weight of IL and nominal training generally improves other realism metrics as expected (Table 3). However, RTR is not particularly sensitive; many configurations are within noise and all configurations dominate the baseline Pareto frontier.

| | | Nominal | | Long-tail |
|---|---|---|---|---|
| $\lambda$ | $\alpha$ | Col. (%) | FDE (m) | Col. (%) |
| 0.0 | 0.0 | $0.89 \pm 0.39$ | $\mathbf{4.50 \pm 0.24}$ | $12.13 \pm 2.44$ |
| 1.0 | 0.5 | $0.38 \pm 0.20$ | $5.50 \pm 0.24$ | $3.61 \pm 1.35$ |
| 5.0 | 0.5 | $0.38 \pm 0.20$ | $5.16 \pm 0.28$ | $3.61 \pm 1.35$ |
| 10 | 0.5 | $0.52 \pm 0.17$ | $5.10 \pm 0.20$ | $3.82 \pm 1.12$ |
| 5.0 | 0.3 | $\mathbf{0.35 \pm 0.18}$ | $5.20 \pm 0.21$ | $4.12 \pm 0.90$ |
| 5.0 | 0.7 | $0.56 \pm 0.21$ | $6.10 \pm 0.23$ | $\mathbf{3.51 \pm 1.32}$ |

Table 3: Balancing realism and infraction avoidance.

**Imitation learning signal:** We consider the alternative of using a frozen pretrained IL policy as regularization [44] instead of our approach of using offline data. A frozen policy potentially provides more accurate closed-loop supervision, as a Euclidian-based distance loss with demonstration data may be inaccurate if the rollout has diverged. We evaluate two baselines: KL Reward and KL Loss, where the KL between the current and frozen policy is added to the reward or loss respectively. Our results in Table 4 show that using demonstration data is still the most performant, suggesting that the inaccuracy from an imperfect IL policy is larger than that of using a distance-based loss.

| | Nominal | | Long-tail |
|---|---|---|---|
| Method | Col. (%) | FDE (m) | Col. (%) |
| KL-L | $0.42 \pm 0.20$ | $25.68 \pm 1.14$ | $5.08 \pm 1.21$ |
| KL-R | $\mathbf{0.38 \pm 0.22}$ | $15.19 \pm 0.99$ | $4.97 \pm 1.29$ |
| RTR | $\mathbf{0.38 \pm 0.20}$ | $\mathbf{5.16 \pm 0.28}$ | $\mathbf{3.61 \pm 1.35}$ |

Table 4: Comparing different alternatives to our proposed IL loss.

## 5 Conclusion and Limitations

We have presented RTR, a method for learning realistic traffic agents with closed-loop IL+RL using both real-world logs and procedurally generated long-tail scenarios. While we have shown substantial improvements over baselines in simulation realism and downstream tasks, we recognize some existing limitations. Firstly, while using logical scenarios as a framework for procedural generation exploits human prior knowledge and is currently an industry standard, manually designing scenarios can be a difficult process, and ensuring an adequate coverage of all possible scenarios is an open problem. Exploring automated alternatives like adversarial approaches to scenario generation would be an interesting future direction. Secondly, while we have explored the downstream task of generating an offline dataset to train prediction models, other applications like training and testing the entire autonomy stack end-to-end in closed-loop is a promising future direction.

**Acknowledgments**

The authors would like to thank Wenyuan Zeng for their insightful discussions throughout the project. The authors would also like to thank the anonymous reviewers for their helpful comments and suggestions to improve the paper.

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

# A  Additional Results

**Metrics:**  In order to measure the realism of our traffic models, we use a set of metrics which evaluate both the traffic models' ability to match human demonstration data in the nominal scenarios and avoid infractions in both nominal and simulated long-tail scenarios.

- *Reconstruction:* In nominal scenarios where expert demonstrations exist, we consider a set of metrics which evaluate how close a traffic model's simulation is to the real world conditioned on the same initial condition. We measure the final displacement error (**FDE**) [65], defined as the L2 distance between an agent's position in a simulated scenario vs the ground truth scenario after 5s. We also measure the along-track error (**ATE**) and cross-track error (**CTE**) of an agent's simulated position projected onto the ground truth trajectory. This decomposition disentangles speed variability and lateral deviations respectively.

- *Distributional*: While reconstruction metrics compare pairs of real and simulated logs, we can compute distributional similarity metrics as an additional method to gauge realism. We compute the Jensen-Shannon Divergence (**JSD**) [31] between histograms of scenario features to compute their distributional similarity. Features include agent kinematics like acceleration and speed, pairwise agent interactions like distance to lead vehicle, and map interactions like lateral deviation from lane centerline.

- *Infraction Rate*: Finally, we measure the rate of traffic infractions made by agents controlled by a traffic model. Similar to prior work [30], we measure percentage of agents that end up in **collision** or drive **off-road**. As this metric does not require ground truth scenarios for pairing or computing statistics, it can be used in simulated long-tail scenarios that do not have ground truth.

**Comparison to state-of-the-art:**  In our main paper, we presented select results from our comparison to state-of-the-art traffic models on both nominal and long-tail scenarios. Here, we include additional tradeoff plots for all metrics in Figure 9. We also include a table of detailed metrics for all methods in Table 5. Building on our observations in the main paper, we see that RTR outperforms and expands the existing Pareto frontier on all metrics and scenario sets. IL methods achieve strong reconstruction/distributional realism metrics but suffer from high infraction rates, while RL methods attain the opposite. RTR achieves the best of both worlds—a testament to its ability to learn human-like driving while avoiding unrealistic traffic infractions.

**Long-Tail Scenarios:**  In our main paper, we evaluated our approach of using procedurally generated long-tail scenarios against the alternative of mining hard scenarios from data. Here, we include additional tradeoff plots for all metrics in Figure 10, with the detailed metrics in Table 6. We see that training on both nominal and long-tail scenarios outperforms the alternatives in most cases.

In addition, we present a slightly different view of Figure 3 of the main paper where the y-axis is in the same scale in Figure 11. This view highlights the difference in difficulty between the different scenario sets.

**Downstream Experiment:**  We provide additional details for our downstream experiment in Section 4.2. The evaluation data used is an additional 118 snippets held out from the nominal dataset. The hyperparameters for training the prediction model (model size, number of epochs, learning rate schedule) were tuned on the training split of the nominal dataset and kept fixed and constant when training on the datasets generated by the methods in Table 2 in order to be fair. The synthetic dataset for each method is also generated using the same 589 initial conditions to be fair. We report the minimum over modes for our multimodal prediction model. We use 4 separate checkpoints to compute the uncertainty estimates.

**Distributional Realism:**  In Figure 12, we include additional plots showing the histograms used to compute JSD distributional realism metrics on the nominal scenario set. We can see that RL methods (RL, RL-Shaped, and BC + RL) struggle to capture human-like driving, particularly in

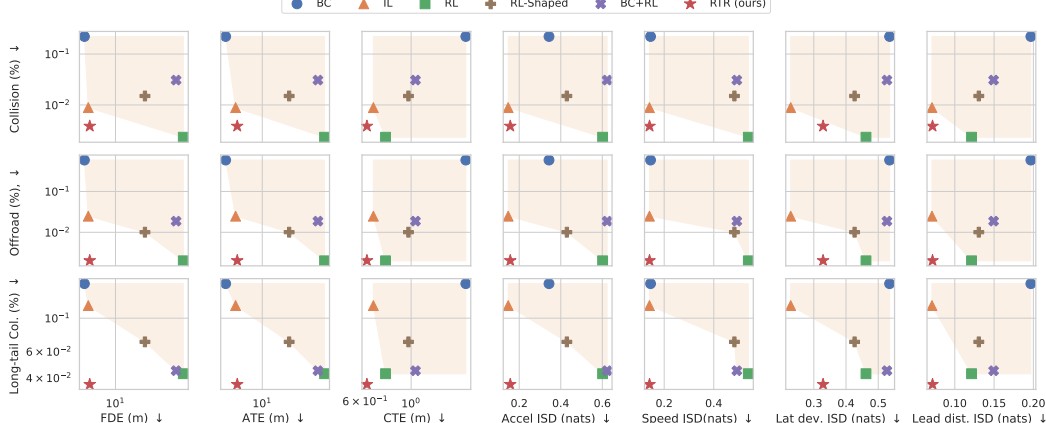

Figure 9: Additional plots comparing infraction / realism tradeoff of RTR compared to baseline models. We see that RTR outperforms and expands the existing Pareto frontier for all metrics.

| | Infraction (%) | | Reconstruction (m) | | | JSD (nats) | | | | LT-Inf. (%) |
|---|---|---|---|---|---|---|---|---|---|---|
| Method | Col. | Off Rd. | FDE | ATE | CTE | Acc. | Spd. | Lat. | Ld. | Col. |
| BC | $22.13 \pm 1.32$ | $58.68 \pm 2.18$ | **4.50 ± 0.24** | **3.60 ± 0.20** | $1.84 \pm 0.17$ | 0.34 | 0.54 | **0.14** | 0.20 | $17.00 \pm 2.91$ |
| IL | $0.89 \pm 0.39$ | $2.48 \pm 0.36$ | $4.98 \pm 0.23$ | $4.75 \pm 0.23$ | $0.66 \pm 0.05$ | **0.15** | **0.23** | **0.14** | **0.07** | $12.13 \pm 2.44$ |
| RL | **0.23 ± 0.17** | $0.20 \pm 0.13$ | $56.92 \pm 0.87$ | $56.91 \pm 1.91$ | $0.75 \pm 0.08$ | 0.60 | 0.46 | 0.54 | 0.12 | $4.26 \pm 1.30$ |
| RL-Shp. | $1.50 \pm 0.36$ | $1.01 \pm 0.29$ | $21.29 \pm 1.13$ | $21.17 \pm 1.12$ | $0.97 \pm 0.05$ | 0.43 | 0.43 | 0.48 | 0.13 | $6.95 \pm 1.81$ |
| BC+RL | $3.08 \pm 0.49$ | $1.88 \pm 0.32$ | $47.30 \pm 0.49$ | $47.26 \pm 0.50$ | $1.05 \pm 0.09$ | 0.62 | 0.53 | 0.49 | 0.15 | $4.46 \pm 1.31$ |
| RTR | $0.38 \pm 0.20$ | **0.20 ± 0.10** | $5.16 \pm 0.28$ | $4.97 \pm 0.28$ | **0.61 ± 0.04** | 0.16 | 0.33 | **0.14** | **0.07** | **3.61 ± 1.35** |

Table 5: Detailed breakdown of metrics. Metrics on the left (resp. right) are computed on nominal scenarios (resp. long-tail scenarios). IL methods achieve strong reconstruction/distributional realism metrics but suffer from high infraction rates, while RL methods attain the opposite. RTR achieves the best of both worlds, with high reconstruction/distributional realism and low infraction rates.

speed and acceleration JSD where the RL methods tend to brake more often than humans. BC exhibits slightly better results overall, but it has worse map interaction reasoning due to distribution shift from compounding errors. In contrast, RTR captures human-like driving significantly better, closely matching IL in distributional realism while also improving on its infraction rate as seen in other results.

**Qualitative Results:** We include qualitative results comparing RTR against the baselines Figures 13, 14, 15, and 16. Across fork, merge, and long-tail scenarios, we see that RTR exhibits the greatest realism of the competing methods.

# B  Learning

## B.1  Loss Derivation

In this section, we will provide more details on the loss derivation using the Lagrangian. Recall that we begin with the following optimization problem

$$\underset{\pi}{\arg\min} \quad D_{\mathrm{KL}}\left(P^{\pi}(\tau) \,\|\, P^{E}(\tau)\right)$$
$$\text{s.t.} \quad \mathbb{E}_{P^{\pi}}\left[R(\tau)\right] \geq 0 \tag{8}$$

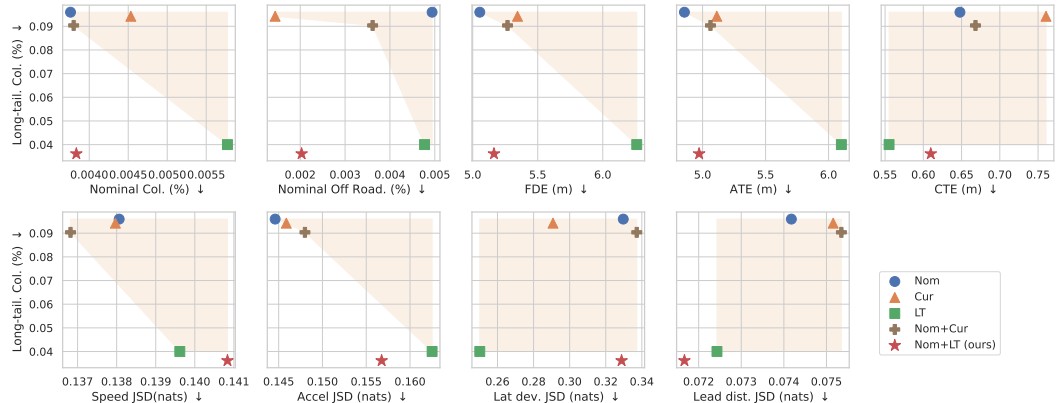

Figure 10: Additional plots showing the tradeoff between infraction rate on the long-tail set and other realism metrics on the nominal set, for models trained on different scenario sets. We see that for most metrics, training on both nominal and long-tail scenarios obtain the best tradeoff.

| | Infraction (%) | | Reconstruction (m) | | | JSD (nats) | | | | LT-Inf. (%) |
|---|---|---|---|---|---|---|---|---|---|---|
| Train | Col. | Off Rd. | FDE | ATE | CTE | Acc. | Spd. | Lat. | Ld. | Col. |
| Nominal | **0.38 ± 0.20** | 0.49 ± 0.17 | **5.05 ± 0.25** | **4.86 ± 0.24** | 0.65 ± 0.04 | 0.14 | 0.14 | 0.33 | **0.07** | 9.60 ± 2.17 |
| Curated | 0.45 ± 0.21 | **0.14 ± 0.08** | 5.34 ± 0.23 | 5.11 ± 0.23 | 0.76 ± 0.05 | **0.15** | 0.14 | 0.29 | 0.08 | 9.42 ± 2.17 |
| Long-tail | 0.58 ± 0.23 | 0.48 ± 0.16 | 6.26 ± 0.38 | 6.10 ± 0.38 | 0.56 ± 0.04 | 0.16 | 0.14 | **0.25** | **0.07** | 4.00 ± 1.37 |
| Nom. + Cur | **0.38 ± 0.20** | 0.30 ± 0.11 | 5.27 ± 0.24 | 5.06 ± 0.30 | 0.67 ± 0.05 | 0.15 | 0.14 | 0.34 | 0.08 | 9.04 ± 2.14 |
| Nom. + LT | **0.38 ± 0.20** | 0.20 ± 0.10 | 5.16 ± 0.28 | 4.97 ± 0.28 | **0.61 ± 0.04** | 0.16 | 0.14 | 0.33 | **0.07** | **3.61 ± 1.35** |

Table 6: Detailed breakdown of realism and infraction metrics for training on different scenario sets.

We form the Lagrangian of the optimization problem

$$\mathcal{L}(\pi, \lambda) = D_{\mathrm{KL}}\left(P^{\pi}(\tau) \parallel P^{E}(\tau)\right) + \lambda \mathbb{E}_{P^{\pi}}\left[R(\tau)\right] \tag{9}$$

$$= \mathbb{E}_{P^{\pi}}\left[\log \frac{P^{\pi}(\tau)}{P^{E}(\tau)} - \lambda R(\tau)\right] \tag{10}$$

$$= \mathbb{E}_{P^{\pi}}\left[-\log P^{E}(\tau) - \lambda R(\tau)\right] - H(\pi). \tag{11}$$

where $\lambda$ is a Lagrangian multiplier and

$$H(\pi) = -\mathbb{E}_{P^{\pi}}\left[\log P^{\pi}(\tau)\right] \tag{12}$$

$$= -\mathbb{E}_{P^{\pi}}\left[\rho_0(\boldsymbol{s}_0) \sum_{t=0}^{T-1} \log \pi(\boldsymbol{a}^t | \boldsymbol{s}^t)\right]. \tag{13}$$

under deterministic dynamics is the causal entropy [26]. Using the Lagragian, the optimization problem is converted to an unconstrained problem

$$\pi^{\star} = \arg\min_{\pi}\max_{\lambda} \mathcal{L}(\pi, \lambda). \tag{14}$$

Equation 14 can be optimized in a number of ways, such as iteratively solving the inner maximization over $\lambda$ and outer minimization over $\pi$. We take a simplified approximate approach where we simply set $\lambda_{\mathrm{fixed}} \geq 0$ as a hyperparameter, leading to what is ultimately a relaxed constraint or penalty method.

$$\pi^* \approx \arg\min_{\pi} \mathbb{E}_{P^{\pi}}\left[-\log P^{E}(\tau) - \lambda_{\mathrm{fixed}} R(\tau)\right] - H(\pi) \tag{15}$$

The causal entropy term is included as an entropy regularization term in some learning algorithms such as PPO [36]. In practice, we found that it was not necessary to include.

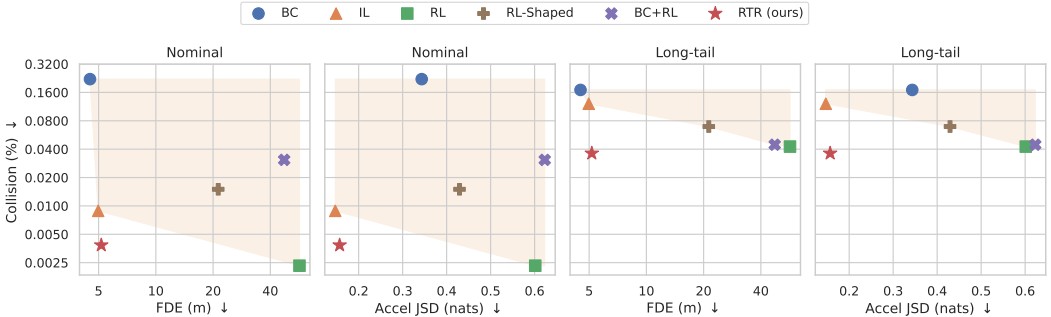

Figure 11: Alternative view of Figure 3, where now the y-axis is on the same scale across the different scenario sets. We see that in the Long-tail scenario set is significantly harder than the nominal set.

## B.2 Imitation Learning Loss

Recall that the imitation learning component of the loss is given as

$$\mathcal{L}^{\mathrm{IL}} = \mathbb{E}_{\tau^E \sim D} \left[ \mathbb{E}_{\tau \sim P^\pi(\cdot|\boldsymbol{s}_0^E)} \left[ D(\tau^E, \tau) \right] \right] \tag{16}$$

$$= E_{(\boldsymbol{s}_0^E, \ldots, \boldsymbol{s}_T^E) \sim D} \left[ \sum_{t=1}^{T} d(\boldsymbol{s}_t^E, \tilde{\boldsymbol{s}}_t) \right] \tag{17}$$

where

$$\tilde{\boldsymbol{a}}_t \sim \pi(\boldsymbol{a}|\tilde{\boldsymbol{s}}_t) \tag{18}$$

$$\tilde{\boldsymbol{s}}_{t+1} = \tilde{\boldsymbol{s}}_t + f(\tilde{\boldsymbol{s}}_t, \tilde{\boldsymbol{a}}_t)\mathrm{d}t. \tag{19}$$

Because the dynamics function $f$ as described in Section B.7 is differentiable, Equation 17 completely differentiable using the reparameterization trick [60] when sampling from the policy. To compute the inner expectation in Equation 16, we simply sample a single rollout. In practice, we found that directly using the mean without sampling is also sufficient.

## B.3 Reward Function

**Sparse reward:** Recall that we use the following reward function

$$R^{(i)}(\boldsymbol{s}, a^{(i)}) = \begin{cases} -1 & \text{if an infraction occurs} \\ 0 & \text{otherwise.} \end{cases} \tag{20}$$

In our experiments, we consider collisions events and driving off-road as infractions. Collisions are computed by checking for overlap between the bounding boxes of agents. Off-road is computed by checking if an agent's bounding box still intersects with the road polygon.

**Early Termination:** Note that when optimizing the reward, we apply early termination of the scenario in the event of an infraction. We treat infractions as terminal states in the MDP for a few reasons. Regarding collision, it is unclear what the optimal behavior (or recovery) looks like after a collision. Similarly, for driving off-road, the actor is likely in a state that it is physically impossible to recover from the real world, as an off-road event would imply the actor has driven off the shoulder into a divider. Finally, in early experiments, we found that continuing simulation for off-road events (and not modeling any shoulders or dividers, physics of off-road driving, etc.) would slow down training since in early phases the policy would drive off-road very early and very severely with no hope of recovering. Resetting in this case prevents wasted simulation in very out-of-distribution states where the policy is completely off the map, etc.

**Shaped reward:**  For the RL-Shaped baseline, use the same reward in Equation 20 with an additional term which encourages driving at the speed limit.

$$R_{shaped}^{(i)}(\boldsymbol{s}, a^{(i)}) = R^{(i)}(\boldsymbol{s}, a^{(i)}) + 0.5(C - \delta)/C \tag{21}$$

where $\delta = \text{abs}(\text{velocity} - \text{speed limit})$ and $C = 30$. For the shaped reward, we additionally terminate the episode if $\delta \geq C$.

## B.4   Reinforcement Learning Loss

We describe our factorized approach to multiagent PPO [36] in more detail. Starting off we compute a per-agent probability ratio.

$$r^{(i)} = \frac{\pi(a^{(i)}|\boldsymbol{s})}{\pi_{\text{old}}(a^{(i)}|\boldsymbol{s})}. \tag{22}$$

Our centralized value-function uses the same architecture as our policy, and computes per-agent value estimates $\hat{V}^{(i)}(\boldsymbol{s})$. Details of the architecture are found in Section B.6. The value model is trained using per-agent value targets, which are computed with per-agent rewards $R_t^{(i)} = R^{(i)}(\boldsymbol{s}_t, a_t^{(i)})$

$$\mathcal{L}^{\text{value}} = \sum_i^N (\hat{V}^{(i)} - V^{(i)})^2 \tag{23}$$

$$V^{(i)} = \sum_{t=0}^T \gamma^t R_t^{(i)} \tag{24}$$

We can obtain a per-agent GAE using the value model as well,

$$A^{(i)} = \text{GAE}(R_0^{(i)}, \ldots, R_{T-1}^{(i)}, \hat{V}^{(i)}(\boldsymbol{s}_T)) \tag{25}$$

The PPO policy loss is simply the sum of per-agent PPO loss,

$$\mathcal{L}^{\text{policy}} = \sum_{i=1}^N \min(r^{(i)}A^{(i)}, \text{clip}(r^{(i)}, 1 - \epsilon, 1 + \epsilon)A^{(i)}) \tag{26}$$

Finally, the overall loss is the sum of the policy and value learning loss.

$$\mathcal{L}^{\text{RL}} = \mathcal{L}^{\text{policy}} + \mathcal{L}^{\text{value}} \tag{27}$$

## B.5   Input Parameterization

**Agent history:**  Following [62], we adopt an viewpoint invariant representation of an agent's past trajectory. We encode the past trajectory as a sequence of pair-wise relative positional encodings between the past waypoints and the current pose. Each relative positional encoding consists of the sine and cosine of distance and heading difference of a pair of poses. See [62] for details.

**Lane graph:**  To construct our lane graph representation $G = (V, E)$, We first obtain the lane graph nodes by discretizing centerlines in the high-definition (HD) map into lane segments every 10m. We use length, width, curvature, speed limit, and lane boundary type (e.g., solid, dashed) as node features. Following [66], we then connect nodes with 4 different relationships: successors, predecessors, left and right neighbors.

## B.6   Model Architecture

Briefly, the RTR model architecture is composed of three main building blocks: (1) context encoders for embedding lane graph and agent history inputs; (2) interaction module for capturing scene-level interaction; and (3a) action decoder for parameterizing the per-agent policy and (3b) value decoder for the value model. Note that the policy model and the value model use the same architecture, but are trained completely separately and do not share any parameters. Early experiments found that not sharing parameters resulted in more stable training – we hypothesize that this is likely because this approach prevents updates to the policy from interfering with the value function, and vice versa.

**History encoder:** The *history encoder* consists of a 1D residual neural network (ResNet) followed by a gated recurrent unit (GRU) that extracts agent features $h_a^{(i)} = f(s^{(i)})$ from a sliding window of past agent states $s$. Intuitively, the 1D CNN captures local temporal patterns, and the GRU aggregates them into a global feature.

**Lane graph encoder:** The *lane graph encoder* is a graph convolutional network (GCN) [66] that extracts map features $h_m = g(m)$ from a given lane-graph $G$ of map $m$. We use hidden channel dimensions of [128, 128, 128, 128], layer normalization (LN), and max pooling aggregation.

**Interaction module:** To model scene-level interaction (i.e., agent-to-agent, agent-to-map, and map-to-map), we build a heterogeneous spatial graph $G'$ by adding agent nodes to the original lane graph $G$. Besides the original lane graph edges, we connect agent nodes to their closest lane graph nodes. All agent nodes are also fully connected to each other. We use a *scene encoder* parameterized by a heterogeneous graph neural network (HeteroGNN) [62] to process map features and agent features into fused features,

$$\{h^{(1)}, \ldots h^{(N)}\} = \text{HeteroGNN}(\{h_a^{(1)}, \ldots, h_a^{(N)}\}, h_m). \tag{28}$$

These fused features are then provided as input to the decoder.

**Action decoder:** Finally, we pass the fused features into a 4-layer MLP with hidden dimensions [128, 128, 128] to predict agent's acceleration and steering angle distributions (parameterized as Normals).

$$(\mu^{(i)}, \sigma^{(i)}) = \text{MLP}(h^{(i)}) \tag{29}$$

$$\pi(a^{(i)}|\mathbf{s}) = \mathcal{N}(\mu^{(i)}, \sigma^{(i)}) \tag{30}$$

**Value decoder:** For the value model, a 4-layer MLP instead regresses a single scalar value representing the value

$$\hat{V}^{(i)} = \text{MLP}_{\text{value}}\left(h_{\text{value}}^{(i)}\right). \tag{31}$$

### B.7 Kinematic Bicycle Model

We use a kinematic bicycle model [59] for our environment dynamics. The bicycle model state is given as

$$s = (x, y, \theta, v) \tag{32}$$

where $x, y$ is the position of the center of the rear axel, $\theta$ is the yaw, and $v$ is the velocity. The bicycle model actions are

$$a = (u, \phi) \tag{33}$$

where $u$ is the acceleration, and $\phi$ is the steering angle. The dynamics function $\dot{s} = f(s, a)$ is then defined as

$$\dot{x} = v \cos(\theta) \tag{34}$$

$$\dot{y} = v \sin(\theta) \tag{35}$$

$$\dot{\theta} = \frac{v}{L} \tan(\phi) \tag{36}$$

$$\dot{v} = u \tag{37}$$

where $L$ is wheelbase length, i.e. the distance between the rear and front axel. We can use a simple finite difference approach to computing the next state

$$s_{t+1} = s_t + f(s_t, a_t)\mathrm{d}t \tag{38}$$

where $\mathrm{d}t$ is chosen to be 0.5 seconds in practice. We can apply the bicycle model to each agent individually to obtain the joint state dynamics function.

**Algorithm 1** RTR Closed-loop Learning

---

1: **for** $n = 1, \cdots, N$ **do**
2:      Set $\mathcal{L}^{\text{IL}} \leftarrow 0$.
3:      Set $\mathcal{L}^{\text{RL}} \leftarrow 0$.
4:      **for** $k = 1, \cdots, K$ **do**
5:          Sample initial state $\boldsymbol{s}_0 \sim (1 - \alpha)\rho_0^D + \alpha\rho_0^S$.
6:          Generate trajectory $\tau$ using policy $\pi_\theta(\boldsymbol{a}|\boldsymbol{s})$ from Equation 3 and simulator.
7:          Compute $\mathcal{L}^{\text{RL}} \leftarrow \mathcal{L}^{\text{RL}} - R(\tau)$.
8:          **if** initial state of $\boldsymbol{s}_0$ is from Nominal Dataset **then**
9:              $\mathcal{L}^{\text{IL}} \leftarrow \mathcal{L}^{\text{IL}} + D(\tau^E, \tau)$.
10:          **end if**
11:      **end for**
12:      Compute $g^{\text{RL}} \leftarrow \nabla_\theta \frac{\lambda}{K} \mathcal{L}^{\text{RL}}$ using our factorized PPO.
13:      Compute $g^{\text{IL}} \leftarrow \nabla_\theta \frac{1}{K} \mathcal{L}^{\text{IL}}$ using BPTT.
14:      Update $\theta$ with $g^{\text{RL}} + g^{\text{IL}}$ using AdamW.
15: **end for**

---

## B.8 Training Details

We use AdamW [67] as our optimizer, and decay the learning rate by a factor of $0.2$ every 3 epochs, and train for a total of 10 epochs. We provide additional training hyperparameters in Table 7. Our overall learning process is summarized in Algorithm 1.

| Hyperparameter | Value |
|---|---|
| IL minibatch size | 32 |
| PPO batch size | 192 |
| PPO minibatch size | 32 |
| PPO num epochs | 1 |
| PPO clip | 0.2 |
| Discount factor | 0.79 |
| Learning rate | 0.00001 |
| Weight decay | 0.0001 |
| GAE $\lambda$ | 1.0 |
| Grad clip norm | 1.0 |

Table 7: Training hyperparameters

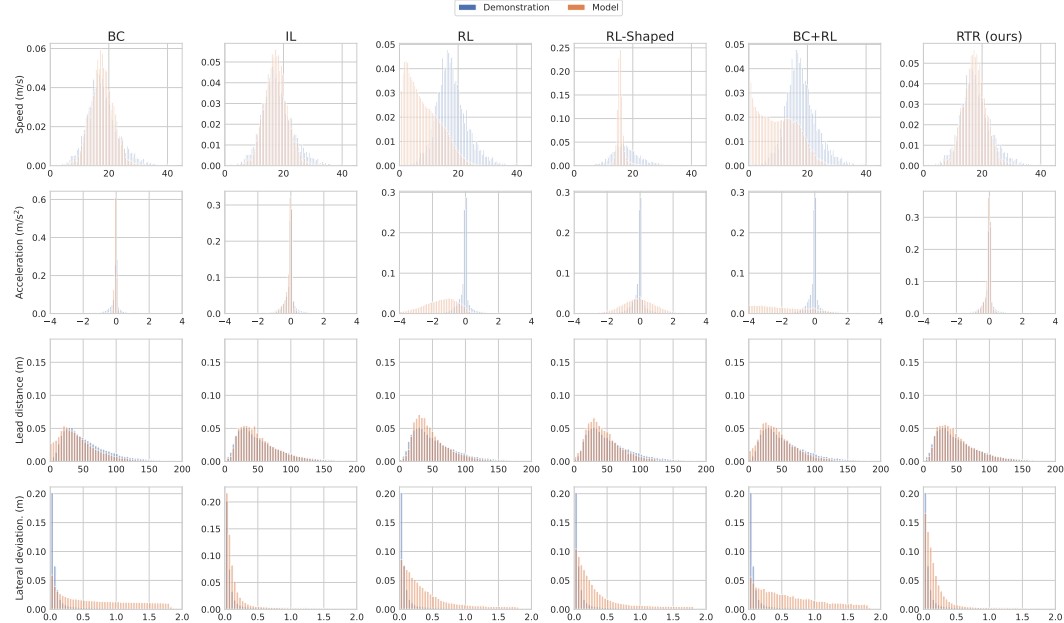

Figure 12: Histograms of scenario features for all methods used to compute JSD distributional realism metrics. We see that BC and RL methods often struggle with capturing the data distribution compared to IL and RTR. Notably, RTR closely matches IL performance in distributional realism, while greatly improving infraction rate as seen in other results.

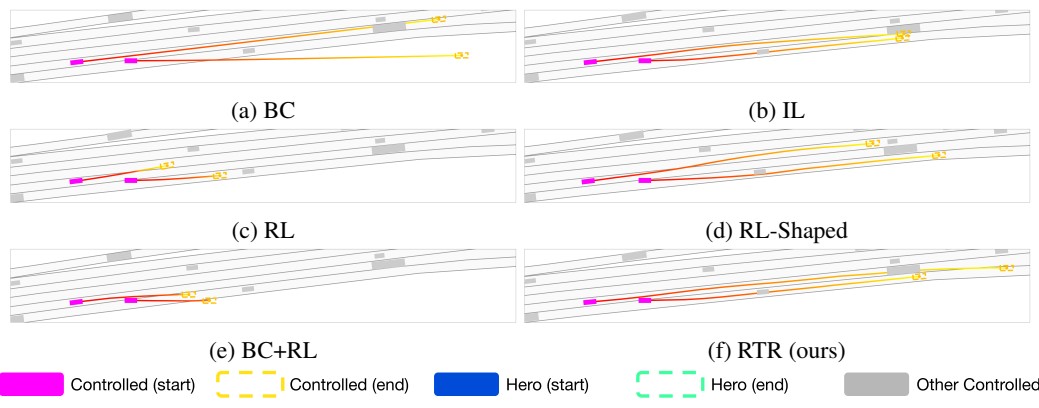

Figure 13: Qualitative results on a fork scenario. BC drives off the road, IL results in a collision while RL and BC+RL slow down. RL-Shaped drives straight and loses the interesting lane change behavior.

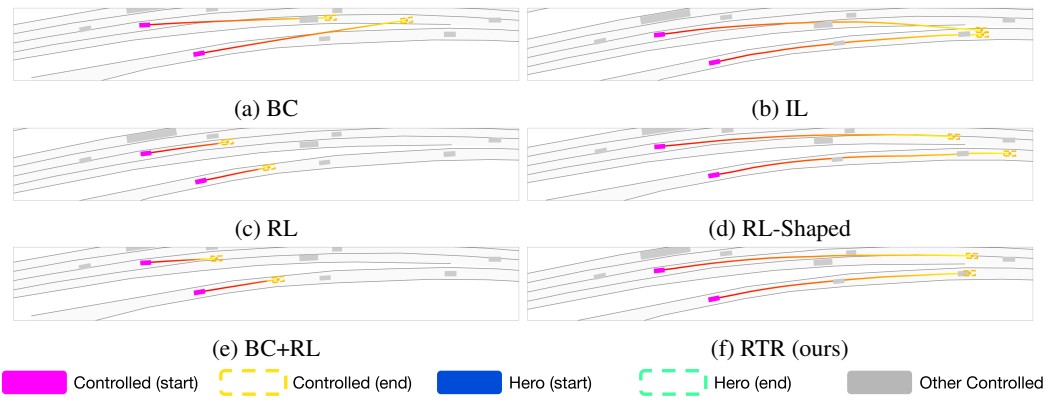

Figure 14: Qualitative results on a merge scenario. We see that RL methods slow down unrealistically. IL results in a collision while RTR maintains realism.

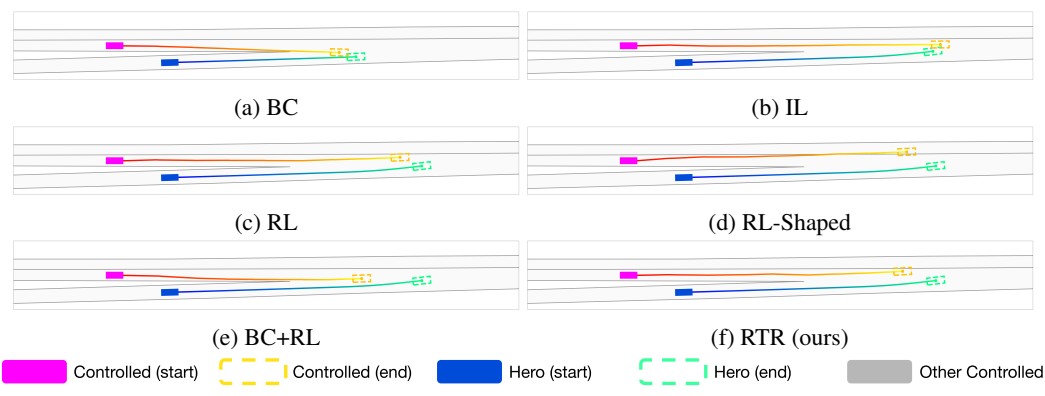

Figure 15: Qualitative results on procedurally generated merge scenario. IL and BC result in a collision. RTR maintains realism.

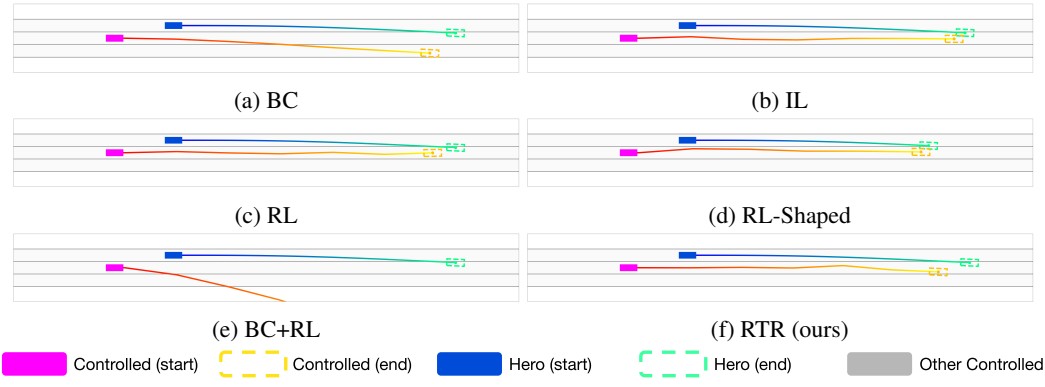

Figure 16: Qualitative results on a procedurally generated cut-in scenario. BC+RL drives off the road, while IL and RL-shaped result in a collision. RTR maintains realism.

