# OpenReview forum: "Learning Realistic Trafﬁc Agents in Closed-loop"
_robot-learning.org/CoRL/2023/Conference — CoRL 2023 Poster_

### Official Review · Reviewer_F38n · 2023-07-17

**Confidence:** 4
**Originality:** Fair
**Technical Quality:** Fair
**Clarity Of Presentation:** Good
**Impact:** 3

**Recommendation:**

Weak Reject: I recommend rejecting the paper, but will not argue for my recommendation if the majority of other reviewers have a different opinion.

**Review:**

There are some suggestions as follows:

1. For the motivation and introduction, there is no doubt that the closed-loop is beneficial for modeling realistic behaviors of traffic agents. An existing work [1] also investigates this problem using the IL+RL framework with a closed-loop simulator, and the motivation and differences should be given.

2. For scenario generation, another important attribute aside from being realistic is being controllable [2, 3]. However, this part is included in the proposed RTR.

3. The biggest problem with RL-based closed loops is scaling and training efficiency. I guess it is one reason why the experiments are carried out on a small dataset with relatively simple highway scenarios. The open-sourced WOMD dataset with urban scenarios should be considered, and stronger baselines should be compared.

4. How many agents are controlled for the nominal and long-tail scenarios? I see only two agents are controlled in the supplementary material. How about the performances with the increasing number of sim agents?

5. The details about the downstream prediction are not given such as the evaluation data. Is it an open-sourced benchmark? I'm not sure about the credibility due to too many missing details.

[1] TrajGEN: Generating realistic and diverse trajectories with reactive and feasible agent behaviors for autonomous driving IEEE Transactions on Intelligent Transportation Systems, 2022
[2] Guided Conditional Diffusion for Controllable Traffic Simulation, arXiv, 2022.
[3] MixSim: A Hierarchical Framework for Mixed Reality Traffic Simulation, CVPR, 2023.
[4] TrafficGen: Learning to Generate Diverse and Realistic Traffic Scenarios, ICRA, 2023.

**Quality Of The Limitations Section:**

Limitations are not well addressed

**Questions For Rebuttal:**

How about the  scaling and training efficiency of the proposed RTR?
How about the performances with the increasing number of sim agents?

**Robotics Focus:**

Highly relevant to robotics but no hardware experiments

**Summary Of Paper:**

This paper investigates the traffic agent behavior modeling task using a joint IL and RL approach in a closed-loop environment. The reward for RL is designed with a sparse signal. Both nominal and long-tail scenarios are generated using the proposed methods. Further, the effectiveness of scenario generation is given for the downstream prediction metrics.

**Summary Of Recommendation:**

This paper investigates the traffic agent behavior modeling task which is  important  for high-level autonomous driving. However, the motivation and the experiments are both inadequate and incomplete.

---

> ### Author Response · Authors · 2023-08-10
> **Response to Reviewer F38n (1/2)**
>
> We thank the reviewer for the valuable comments, and respond below.
>
> **W1:**
> > For the motivation and introduction, there is no doubt that the closed-loop is beneficial for modeling realistic behaviors of traffic agents. An existing work [1] also investigates this problem using the IL+RL framework with a closed-loop simulator, and the motivation and differences should be given.
>
> We agree with the reviewer that closed-loop is crucial for modeling realistic behaviors, and thank the reviewer for bringing this relevant work to our attention. As the reviewer mentioned, [1] shares some similarities with our work by using an IL+RL framework with closed-loop simulation. In more detail, [1] uses a two-stage approach where stage 1 consists of an open-loop prediction IL objective and stage 2 uses a closed-loop RL approach. Specifically, stage 1 can suffer from distribution shift and compounding error when encountering suboptimal states, producing suboptimal tracking trajectories for stage 2.  In fact, many prior works have similarly used an open-loop IL and closed-loop RL approach [6,7]. Fundamentally, this mismatch leads to the issue where the open-loop IL objective is only applied to expert states, leaving the model susceptible to compounding error and distribution shift during closed-loop evaluation, despite having been trained with closed-loop RL. Our experiments with our BC+RL baseline provide evidence for this. Thus one contribution of our work is a holistic closed-loop learning objective to match expert demonstrations under a traffic compliance constraint, naturally giving rise to a joint closed-loop IL+RL approach which we show to be much more robust to distribution shift.
>
> Secondly, our work shows that learning using additionally procedurally generated long-tail solutions results in more realistic and robust traffic simulation policies. While [1] uses additional safety-critical scenarios to evaluate the safety of an autonomy policy, our work explicitly shows that training on procedurally generated scenarios results in more generalizable policies (Figure 6, Table 1) that are not just more conservative but also more realistic (Figure 3, Table 2). We will add this discussion to the related work in the camera ready.
>
> **W2:**
> > For scenario generation, another important attribute aside from being realistic is being controllable [2, 3]. However, this part is included in the proposed RTR.
>
> We agree with the reviewer that both realism and controllability are important factors in traffic simulation. However, we’d like to clarify that RTR focuses on improving realism, and thus focused on the task of simpler task of unconditional simulation. Controllable/conditional traffic simulation also struggles with realism, and many techniques proposed in RTR (holistic closed-loop IL+RL, additional procedurally generated long-tail scenarios for training, etc.) are applicable in an orthogonal fashion. It would also be an interesting line of future work to consider controllability through modifications of RTR, such as unifying route-conditional losses in [3] with goal-conditioned RL, etc.

---

> ### Author Response · Authors · 2023-08-10
> **Response to Reviewer F38n (2/2)**
>
> **W3:**
> > The biggest problem with RL-based closed loops is scaling and training efficiency. I guess it is one reason why the experiments are carried out on a small dataset with relatively simple highway scenarios. The open-sourced WOMD dataset with urban scenarios should be considered, and stronger baselines should be compared.
>
>  We’d like to clearly disentangle the scalability of our method from the experimental setting/baselines we have considered. The experiments in RTR tackle traffic simulation for highway scenarios, which overlap in the challenges faced in urban driving.  To our knowledge, the most commonly used public highway dataset is NGSIM [5], which contains a single 500m stretch of I80 containing a single on ramp recorded for 45 min, and a single 640m stretch of US101 containing a single on and off ramp recorded for 45 min. Our combined nominal and long-tail scenario dataset is much larger, being over 300 minutes, containing 18 different geolocations covering multiple highways across North America  containing several on and off ramps, merges and forks topologies, and additional simulated maps of varying curvatures for the procedurally generated scenarios.  Highway scenarios often require complex negotiations at high speeds, and can be particularly difficult in our long-tail set. Our baselines are also representative of state-of-the-art traffic simulation approaches, with details in L218-225.  Finally, while highway datasets and urban driving datasets cannot be directly compared, we do acknowledge that our highway dataset is smaller than some of larger urban driving datasets (e.g. WOMD). However, it is still comparable to other popular urban datasets in size alone (e.g. NuScenes is 333 minutes, INTERACTIONS is 990 minutes).
>
> Regarding scaling and training efficiency, we find that with our factorized approach towards large-scale multiagent learning, our method is able to train well with a large (30-50) number of actors, and has much better sample efficiency compared to the naive unfactorized approach. Our joint IL+RL approach also accelerates training compared to using pure RL.
>
> **W4:**
> > How many agents are controlled for the nominal and long-tail scenarios? I see only two agents are controlled in the supplementary material. How about the performances with the increasing number of sim agents?
>
> To clarify, for nominal scenarios, all actors are controlled by RTR and for long-tail scenarios, all actors except the hero/scripted actors are controlled by RTR. This varies across scenarios, but averages out to roughly 30, and can peak to roughly 50 when the traffic density is high. In our visualizations (both figures and supplementary video), we highlight scripted actors with blue, and often highlight 2 actors with pink in order to draw the viewer's attention to them, but stress that all other gray actors are also being controlled. We will update the camera ready supplementary material to be more clear.
>
> Our method also scales well with the number of agents from a runtime efficiency perspective, since all computation is batched on the GPU, and our centralized design eliminates overhead from redundant computation as reviewer CEx1 also pointed out.
>
> **W5:**
> > The details about the downstream prediction are not given such as the evaluation data. Is it an open-sourced benchmark? I'm not sure about the credibility due to too many missing details.
>
> The evaluation data used is an additional 118 snippets held out from the nominal dataset described in L204-205 and our response to W3.  The hyperparameters for training the prediction model (model size, number of epochs, learning rate schedule) were tuned on the training split of the nominal dataset and kept fixed and constant when training on the datasets generated by the methods in Table 2 in order to be fair. The synthetic dataset for each method is also generated using the same 589 initial conditions to be fair. We will include these details in the camera ready supplementary.
>
> **References:**
> [1] TrajGEN: Generating realistic and diverse trajectories with reactive and feasible agent behaviors for autonomous driving IEEE Transactions on Intelligent Transportation Systems, 2022
> [2] Guided Conditional Diffusion for Controllable Traffic Simulation, arXiv, 2022.
> [3] MixSim: A Hierarchical Framework for Mixed Reality Traffic Simulation, CVPR, 2023.
> [4] TrafficGen: Learning to Generate Diverse and Realistic Traffic Scenarios, ICRA, 2023.
> [5] John Halkias and James Colyar. NGSIM interstate 80 freeway dataset. US Federal Highway Administration. 2006.
> [6] Lu, Yiren, et al. "Imitation is not enough: Robustifying imitation with reinforcement learning for challenging driving scenarios." arXiv (2022).
> [7] Kamenev, Alexey, et al. "Predictionnet: Real-time joint probabilistic traffic prediction for planning, control, and simulation." 2022 International Conference on Robotics and Automation (ICRA). IEEE, 2022.

---

> > ### Comment · Reviewer_F38n · 2023-08-15
> > **Response**
> >
> > I thank the authors for the detailed reply and revisions. After reading the rebuttal, my evaluation is weak reject. Due to the lack of controllable attribute and validation on a single scenario dataset, I think this manuscript could be improved further.

---

> > > ### Author Response · Authors · 2023-08-16
> > > **Response**
> > >
> > > Thanks the reviewer for their response.
> > >
> > > The focus of RTR was realism, and we evaluated the unconditional simulation task, which is a common and important use case in the industry. However, our method is agnostic to whether we perform unconditional or conditional (ie controllable simulation). For example, RTR can be augmented to be controllable by swapping out the architecture with the route-conditional architecture found in MixSim [3]. Studying controllability is orthogonal, and lack of controllability is not an inherit weakness of our approach.
> > >
> > > We perform validation on the Nominal dataset, in-distribution long tail scenarios, and out-of-distribution long tail scenarios, which we believe extensively capture methods' realism, robustness and generalization ability.

---

> ### Author Response · Authors · 2023-08-15
> **Follow up**
>
> Dear reviewer, as the discussion phase is ending soon, we'd like to make sure if you have any additional concerns that we could discuss or address. Thank you!

---

### Official Review · Reviewer_CEx1 · 2023-07-17

**Confidence:** 5
**Originality:** Good
**Technical Quality:** Good
**Clarity Of Presentation:** Very Good
**Impact:** 3

**Recommendation:**

Weak Accept: I recommend accepting the paper, but will not argue for my recommendation if the majority of other reviewers have a different opinion.

**Review:**

### Strength:
1. The motivation is good. Reactive traffic simulation is important to Autonomous Driving (AD) field, especially considering almost every company uses the recorded and fixed driving log to test their systems.
2. Technically, the method provides some interesting ideas. For example, combining recorded data and synthetic data together to build a mixed dataset enables the policy to solve interaction-rich safety-critical scenarios which are rare and impossible to record in the real world; The IL objective is learned through differential simulation via bicycle model; The centralized policy design encoding the whole context, avoiding computational overhead.
3. Extensive experiments are conducted to demonstrate the effectiveness of the proposed method. Also, downstream tasks prove the potential of the traffic simulation model.

### Suggestions:
1. Some references can be included to provide recent advances in traffic simulation.
>Zhong, Ziyuan, et al. "Guided conditional diffusion for controllable traffic simulation." 2023 IEEE International Conference on Robotics and Automation (ICRA). IEEE, 2023.
Feng, Lan, et al. "Trafficgen: Learning to generate diverse and realistic traffic scenarios." 2023 IEEE International Conference on Robotics and Automation (ICRA). IEEE, 2023.
Rempe, Davis, et al. "Generating useful accident-prone driving scenarios via a learned traffic prior." Proceedings of the IEEE/CVF Conference on Computer Vision and Pattern Recognition. 2022.
Two of them, CTG and STRIVE, are related to safety-critical or long-tail scenario generation based on original scenarios. The other one, TrafficGen, focuses on simulating both initial conditions and subsequent trajectory generation given an empty map. The settings of these papers is not totally as same as RTR, but some overlaps do exist.

2. A improvement of the method in my mind is to exempt the need for estimating the value function. Actually, one can directly optimize the RL objective with a differential simulation. Note I am not saying you should try this right now. I just want to give a potential improvement that can be adopted to make RTR better. With the differentiable bicycle model, the displacements between two frames can serve as dense rewards, and a termination reward can be raised by collision or driving off the road. In this way, both IL and RL would be unified into the RL framework whose goal is to follow a referenced trajectory. Besides, the dense reward signal can enjoy the TD, making the imitation episode-wise, which is not allowed by doing step-wise imitation learning through current IL loss. One feasible method is:
>Mora, Miguel Angel Zamora, et al. "Pods: Policy optimization via differentiable simulation." International Conference on Machine Learning. PMLR, 2021.

3. Consider replacing long-tail with safety-critical, which is more accurate than long-tail and widely used/accepted
4. typo line 241
5. please report std in all tables
6. I think one ablation result should be added where the $\lambda$ is set to zero and thoroughly ablate the RL component. A performance drop in terms of collision/driving off the road can demonstrate the necessity to have RL loss.
7. The conclusion of section 4.2 is that the generated scenarios can be used to train the trajectory prediction model. Indeed, the model trained with driving logs can generate a surrogate training dataset, while the performance of training a prediction model with the surrogate data is supposed to be worse than training the prediction model directly with the recorded scenario. This makes this application not interesting and promising enough. However, this experiment is actually good evidence demonstrating your agent model learns better data distribution and driving behaviors than other baselines according to the evaluation results on the held-out real-world test set. Thus I recommend using this experiment to support your main result: RTR can learn realistic driving behavior.

BTW, I think this model-based motion prediction is an interesting application, as most of the motion prediction methods directly regress the velocity/position of the next several frames. Unlike these methods, RTR applies hard kinematics constraints by using the bicycle model and predicts actions that will be further processed by the model, which may produce better prediction results. Maybe authors can evaluate it on Waymo or nuScenes data to see if it can achieve SOTA.






**Quality Of The Limitations Section:**

Limitations are addressed clearly

**Questions For Rebuttal:**

N/A

**Robotics Focus:**

Relevant but unlikely to deploy to hardware in near future

**Summary Of Paper:**

Simulation is the very first step of deploying self-driving systems into the real world. The key to traffic simulation is ensuring the multi-agent system is realistic and able to react to other vehicles, i.e. ego-car with an autonomous driving system. This paper proposes a new method called RTR to simulate the traffic where simulated vehicles can not only retain the realistic driving behaviors in recorded driving logs but solve long-tail safety-critical scenarios. Extensive experiments are conducted to prove the effectiveness and advantages of RTR with both qualitative and quantitative results.

**Summary Of Recommendation:**

I recommend accepting this paper. But it is still likely to increase or decrease my score depending on the rebuttal and other reviewers' comments.

---

> ### Author Response · Authors · 2023-08-10
> **Response to Reviewer CEx1 (1/2)**
>
> We thank the reviewer for the valuable comments, and respond below.
>
> **Q1:**
> > Some references can be included to provide recent advances in traffic simulation. Two of them, CTG and STRIVE, are related to safety-critical or long-tail scenario generation based on original scenarios. The other one, TrafficGen, focuses on simulating both initial conditions and subsequent trajectory generation given an empty map. The settings of these papers is not totally as same as RTR, but some overlaps do exist.
>
> We thank the reviewer for these references and will include them in the updated related work for the camera ready. CTG [1] is a traffic simulation approach that leverages STL for controllable simulation through guided sampling of diffusion models. This approach is able to reduce infraction rate at test time, but sampling from diffusion models can be expensive. STRIVE [2] generates adversarial scenarios with emphasis on realism and solvability of the scenario. Exploring adversarial approaches like STRIVE to generate training scenarios for RTR (as opposed to our procedurally generated long-tail scenarios) can be an interesting direction for future work. And as the reviewer mentioned, TrafficGen [3] generates scenarios from an empty map as opposed to leveraging initial conditions like prior works. They show that training a naive RL agent with generated data can outperform real data. It would also be interesting to see if these scenarios can be used for RTR.
>
> **Q2:**
> > An improvement of the method in my mind is to exempt the need for estimating the value function. Actually, one can directly optimize the RL objective with a differential simulation. Note I am not saying you should try this right now. I just want to give a potential improvement that can be adopted to make RTR better. With the differentiable bicycle model, the displacements between two frames can serve as dense rewards, and a termination reward can be raised by collision or driving off the road. In this way, both IL and RL would be unified into the RL framework whose goal is to follow a referenced trajectory. Besides, the dense reward signal can enjoy the TD, making the imitation episode-wise, which is not allowed by doing step-wise imitation learning through current IL loss.
>
> We thank the reviewer for the interesting ideas and references. Indeed because we are using a differentiable simulator, it would be interesting to see if the PODS method can be adopted to improve RTR. Note that the current imitation loss is applied to a trajectory with horizon T and optimized with BPTT, so it should be possible to set T to the episode length which would make the imitation episode wise (in practice T is fixed to 5s for simplicity though).
>
> **Q3:**
> > Consider replacing long-tail with safety-critical, which is more accurate than long-tail and widely used/accepted
>
> We thank the reviewer for this suggestion. We have also considered the term safety-critical but found that it may be misleading as not all long-tail scenarios generated are necessarily safety-critical.
>
> **Q4:**
> >typo line 241
>
> Thanks, this will be fixed in the camera ready.
>
> **Q5:**
> > please report std in all tables
>
> Thanks for the suggestion, we will report std in the camera ready pdf.
>
> **Q6:**
> > I think one ablation result should be added where the lambda is set to zero and thoroughly ablate the RL component. A performance drop in terms of collision/driving off the road can demonstrate the necessity to have RL loss.
>
> We’d like to clarify that the IL baseline is this ablation. This baseline uses the same IL loss but has lambda set to zero, and alpha (controlling the proportion of long-tail scenarios) set to zero, since there is no demonstration for those scenarios and so IL cannot be applied. From the results we indeed see that the IL baseline has worse infraction rate compared to RTR, on both nominal and long-tail evaluation. We will update the camera ready to make this more clear.
>
> **Q7**:
> > The conclusion of section 4.2 is that the generated scenarios can be used to train the trajectory prediction model. Indeed, the model trained with driving logs can generate a surrogate training dataset, while the performance of training a prediction model with the surrogate data is supposed to be worse than training the prediction model directly with the recorded scenario. This makes this application not interesting and promising enough. However, this experiment is actually good evidence demonstrating your agent model learns better data distribution and driving behaviors than other baselines according to the evaluation results on the held-out real-world test set. Thus I recommend using this experiment to support your main result: RTR can learn realistic driving behavior.
>
> We thank the reviewer for this suggestion, and we agree that indeed the downstream evaluation can be seen as evidence that RTR-generated data has a lower domain gap compared to baselines. We will update the camera ready with this discussion.

---

> ### Author Response · Authors · 2023-08-10
> **Response to Reviewer CEx1 (2/2)**
>
> **Q8:**
> > BTW, I think this model-based motion prediction is an interesting application, as most of the motion prediction methods directly regress the velocity/position of the next several frames. Unlike these methods, RTR applies hard kinematics constraints by using the bicycle model and predicts actions that will be further processed by the model, which may produce better prediction results. Maybe authors can evaluate it on Waymo or nuScenes data to see if it can achieve SOTA.
>
> We thank the reviewer for their comment and suggestion. We agree that using model-based motion prediction could be an interesting alternative direction compared to directly regressing velocity/position. While it provides a strong inductive bias, it also has its own limitations when applied directly to motion prediction, rather than traffic simulation. For example, initial errors in the estimation of velocity can propagate and result in poor results - this is less of an issue in the traffic simulation setting where we have the ground truth simulation state. Nevertheless, we also agree that evaluating a model-based prediction approach on Waymo or NuScenes is an interesting line of future research.
>
> **References:**
> [1] Zhong, Ziyuan, et al. "Guided conditional diffusion for controllable traffic simulation." 2023 IEEE International Conference on Robotics and Automation (ICRA). IEEE, 2023
> [2] Rempe, Davis, et al. "Generating useful accident-prone driving scenarios via a learned traffic prior." Proceedings of the IEEE/CVF Conference on Computer Vision and Pattern Recognition. 2022.
> [3] Feng, Lan, et al. "Trafficgen: Learning to generate diverse and realistic traffic scenarios." 2023 IEEE International Conference on Robotics and Automation (ICRA). IEEE, 2023.
> [4] Kamenev, Alexey, et al. "Predictionnet: Real-time joint probabilistic traffic prediction for planning, control, and simulation." 2022 International Conference on Robotics and Automation (ICRA). IEEE, 2022.
> [5] TrajGEN: Generating realistic and diverse trajectories with reactive and feasible agent behaviors for autonomous driving IEEE Transactions on Intelligent Transportation Systems, 2022

---

> > ### Comment · Reviewer_CEx1 · 2023-08-12
> > **Thank you for your response**
> >
> > Thank you for the detailed response. My concerns are all addressed. Nice work!

---

### Official Review · Reviewer_ohnK · 2023-07-21

**Confidence:** 3
**Originality:** Very Good
**Technical Quality:** Very Good
**Clarity Of Presentation:** Very Good
**Impact:** 3

**Recommendation:**

Weak Accept: I recommend accepting the paper, but will not argue for my recommendation if the majority of other reviewers have a different opinion.

**Review:**

#### Quality
- The paper is technically sound. The claims are supported by the empirical experimental results. The method is appropriate.

#### Clarity
- The paper is relatively clearly written and well organized.

#### Originality
- Neither the problem nor the method is new. The proposed method is a combination of GAIL (without the discriminator, but instead using a concrete distance function) and standard on-policy RL with factorized rewards in a multi-agent setting. The method is incremental to the previous similar methods such as BC+RL [1].


#### Significance
- The results are interesting and relevant to the field. Others are likely to use the idea.

#### Strengths
- The paper is relevantly well written
- The proposed method is straightforward and effective - it’s likely to be adopted.
- Extensive experiments (both qualitative and quantitative). Ablation results show the effects of a few important parameters.

#### Weaknesses
- The proposed method is a combination of existing well known methods. Technical novelty is limited.
- The experimental results largely follow the similar results found in paper [1].
- Dataset is relatively small (~600 segments in total).
- No related work in long-tail scenario generation. Although the paper mentioned adversarial approaches in the limitation paragraph.


[1] Imitation Is Not Enough: Robustifying Imitation with Reinforcement Learning for Challenging Driving Scenarios


**Quality Of The Limitations Section:**

Limitations are addressed clearly

**Questions For Rebuttal:**

- Any comparisons with other long-tail scenario generation methods? No related work in long-tail scenario generation.
- Could the authors provide the reward definition? How does reward shaping affect the performance?
- The progress is not reported as a metric. I would be curious to know if the method makes sufficient progress while maintaining low infractions and high imitation fidelity.
- Is the hero agent log-following for the nominal scenarios? Have the authors explored any other hero agent policies?



**Robotics Focus:**

Highly relevant to robotics but no hardware experiments

**Summary Of Paper:**

The paper proposed a combined imitation learning (IL) and reinforcement learning (RL) approach, named Reinforcing Traffic Rules (RTR), to learn driving policies for traffic agents in closed-loop. RTR optimizes the IL component by minimizing the sum of distance between the expert states and policy states with huber loss. Given the differentiable transition dynamics, the policy can be directly optimized with the reparameterization trick. The RL component is optimized by standard on-policy methods, i.e., PPO with a factorized multi-agent reward function that penalizes infractions such as collisions and off-roads.

The paper also simulates long-tail scenarios based on logical scenarios to evaluate the out-of-distribution generalization of the RTR method. The paper conducts extensive experiments to demonstrate that the proposed method is effective for both learning realistic driving behavior and being able to avoid driving safety events.



**Summary Of Recommendation:**

The paper proposed a combined imitation learning (IL) and reinforcement learning (RL) approach, named Reinforcing Traffic Rules (RTR), to learn driving policies for traffic agents in closed-loop. RTR optimizes the IL component by minimizing the sum of distance between the expert states and policy states with huber loss. The RL component is optimized by standard on-policy methods, i.e., PPO with a factorized multi-agent reward function that penalizes infractions such as collisions and off-roads. Long-tail scenarios are generated based on logical scenarios to evaluate the out-of-distribution generalization of the method. The paper conducts extensive experiments to demonstrate that the effectiveness of the proposed method.

- The paper is well written and easy to follow.
- Although the technical novelty is limited, the results are relevant and the method is likely to be adopted.

I recommend a weak accept.

---

> ### Author Response · Authors · 2023-08-10
> **Resposne to Reviewer ohnK (1/2)**
>
> We thank the reviewer for the valuable comments, and respond below.
>
> **W1/W2:**
> > Neither the problem nor the method is new. The proposed method is a combination of GAIL (without the discriminator, but instead using a concrete distance function) and standard on-policy RL with factorized rewards in a multi-agent setting. The method is incremental to the previous similar methods such as BC+RL [1]. The proposed method is a combination of existing well known methods. Technical novelty is limited.  The experimental results largely follow the similar results found in paper [1].
>
> While the high level task of traffic simulation is not new, we argue that we have identified new problems present in many related works and offer novel solutions which significantly outperform them. Firstly, [1] and other works in traffic simulation combining IL and RL [4, 5] all take the approach of utilizing an open-loop IL and a closed-loop RL objective. Unfortunately, open loop imitation only provides regularization in expert states. In non-expert states, only the RL signal is present, and there is no imitation objective. As a result, as you unroll the policy, small errors accumulate and push the policy into non-expert states. Here, the policy has never received IL regularization, leading to worse imitation. We demonstrate this failure mode of [1] in our experiments.
> Our solution of formulating a holistic closed-loop learning approach is novel to our  significantly outperforms [1] in our experiments as it is more robust to distribution shift. Our novel factorized + centralized approach to multiagent learning is shown to be crucial to scale to many agents, as shown in Figure 8.
>
> Secondly, we show that rather than relying on extreme large-scale data collection and curation [1,7] leveraging procedurally generated long-tail scenarios can greatly improve the quality and robustness of learned traffic simulation policies. Our Nom+Cur baseline is representative of the curation approach taken in [1], and Figure 7, L267-268 shows that our approach is more effective. Curating safety critical scenarios [1] or adversarially generating them [3, 4] has been studied in the context of autonomy, where the larger emphasis is on safety and conservative driving. Rather than focusing on just more conservative autonomy, our approach results in overall more realistic and human-like traffic simulation policies, which is novel to our knowledge.
>
> **W3:**
> > Dataset is relatively small (~600 segments in total).
>
> The dataset used in RTR is for highway scenarios - to our knowledge, the most commonly used public highway dataset is NGSIM [9], which contains a single 500m stretch of I80 containing a single on ramp recorded for 45 min, and a single 640m stretch of US101 containing a single on ramp and off ramp recorded for 45 min. Our combined nominal and long-tail scenario dataset is larger, being over 300 minutes, containing 18 different geolocations covering multiple highways across North America containing several on ramp, off ramp, merge and fork topologies, and additional simulated maps of varying curvatures for the procedurally generated scenarios. While highway datasets and urban driving datasets cannot be directly compared, we do acknowledge that our highway dataset is smaller than some of larger urban driving datasets (e.g. WOMD). However, it is still comparable to other popular urban datasets (e.g. NuScenes is 333 minutes, INTERACTIONS is 990 minutes).

---

> > ### Comment · Reviewer_ohnK · 2023-08-16
> > **Response to the authors**
> >
> > I would like to thank the authors for the responses. The responses make sense and most of my concerns are addressed.
> >
> > That said, given the current state of the manuscript, my recommendation remains the same as weak accept. I hope the authors improve the draft in the camera-ready version as promised in the responses if the paper gets accepted.

---

> ### Author Response · Authors · 2023-08-10
> **Response to Reviewer ohnK (2/2)**
>
> **W4/Q1:**
> > No related work in long-tail scenario generation. Although the paper mentioned adversarial approaches in the limitation paragraph. Any comparisons with other long-tail scenario generation methods?
>
> We thank the reviewer for this suggestion and will update the related work section in the camera ready to include discussion on long-tail scenario generation. One can categorize approaches 1) data-driven generation, 2) adversarial generation and 3) knowledge-based generation [8]. Purely data driven approaches often involve replaying curated collected logs [1,2]. This however relies on large-scale data collection to curate enough long-tail scenarios, and the resulting simulations may not be realistic if actors are constrained to follow their logged trajectories. Adversarial approaches try to generate scenarios which maximize an adversarial objective [3,4,5]. In order to generate scenarios that are actually useful for training, some realism and solvability considerations are taken [3], but it is still an open problem on designing the adversarial objective to take into account additional factors such as coverage, etc. Finally, knowledge-based generation approaches use human expertise and domain knowledge to guide the scenario generation process. Approaches like [6] and ours leverage human expertise to design logical scenarios that cover a wide range of possibilities and procedurally generate a large variety of scenarios.
>
> While one key advantage of our approach is exploiting synthetically generated long-tail scenarios, the specific generation method used is not the focus. We chose to use a knowledge-based approach as opposed to adversarial approaches due to the latter being more opaque and more difficult to tune. However, a more in-depth study in other approaches, or potentially developing a better automated approach is an interesting line of future work.
>
> **Q2:**
> > Could the authors provide the reward definition? How does reward shaping affect the performance?
>
> Equation 2 in the main paper / Equation 13 in the supplementary is the sparse reward definition that our method and most baselines use, where an infraction is defined as either a collision (an intersection between two vehicle polygons) or off-road (no intersection between a vehicle polygon and road polygon). Equation 14 in the supplementary describes the shaped reward baseline, which includes an additional term to encourage driving at the speed limit. In Figure 3, we see that shaping the reward causes more ‘human-like’ driving at the cost of a regression in infraction-avoidance compared to the sparse reward.
>
> **Q3:**
> > The progress is not reported as a metric. I would be curious to know if the method makes sufficient progress while maintaining low infractions and high imitation fidelity.
>
> While progress is not reported, many of the other metrics give a good indication. Firstly, the standard FDE metrics will show a regression in progress since the demonstration data has good progress. The JSD metrics can also show when progress is negatively impacted. For example, Figure 5 shows the naive policy heavily favors decelerating to avoid collisions, which negatively impact progress.
>
> **Q4:**
> > Is the hero agent log-following for the nominal scenarios? Have the authors explored any other hero agent policies?
>
> For nominal scenarios, there are no hero agents, and all agents are controlled by the policy. Hero agents are only present for the long-tail scenarios, and their behavior varies across scenarios. Exploring hero-agent-augmented nominal scenarios could be an interesting direction for future work. We thank the reviewer for the feedback and will make this more clear in the camera-ready paper.
>
> **References:**
> [1] Lu, Yiren, et al. "Imitation is not enough: Robustifying imitation with reinforcement learning for challenging driving scenarios." arXiv (2022).
> [2] Webb, Nick, et al. "Waymo's safety methodologies and safety readiness determinations." arXiv (2020).
> [3] Rempe, Davis, et al. "Generating useful accident-prone driving scenarios via a learned traffic prior." ICCV 2022.
> [4] Hanselmann, Niklas, et al. "King: Generating safety-critical driving scenarios for robust imitation via kinematics gradients." ECCV 2022.
> [5] Wang, Jingkang, et al. "Advsim: Generating safety-critical scenarios for self-driving vehicles." CVPR 2021.
> [6] Menzel, Till, Gerrit Bagschik, and Markus Maurer. "Scenarios for development, test and validation of automated vehicles."IV 2018.
> [7] Bronstein, Eli, et al. "Embedding Synthetic Off-Policy Experience for Autonomous Driving via Zero-Shot Curricula." CORL 2023.
> [8] Ding, Wenhao, et al. "A survey on safety-critical driving scenario generation—A methodological perspective." IEEE Transactions on Intelligent Transportation Systems (2023).
> [9] John Halkias and James Colyar. NGSIM interstate 80 freeway dataset. US Federal Highway Administration, FHWA-HRT-06-137, Washington, DC, USA, 2006.

---

> ### Author Response · Authors · 2023-08-15
> **Follow up**
>
> Dear reviewer, as the discussion phase is ending soon, we'd like to make sure if you have any additional concerns that we could discuss or address. Thank you!

---

### Official Review · Reviewer_gvqu · 2023-07-24

**Confidence:** 4
**Originality:** Good
**Technical Quality:** Very Good
**Clarity Of Presentation:** Very Good
**Impact:** 3

**Recommendation:**

Weak Accept: I recommend accepting the paper, but will not argue for my recommendation if the majority of other reviewers have a different opinion.

**Review:**

Praise:
- The paper tackles an important problem of improvement of traffic simulation realism.
- The idea of factorizing value estimation together with the policy is interesting.
- The work compares against a wide range of baselines. The findings of ablation studies in this work are important for RL practitioners in AV.
- Demonstration in the improvement of downstream tasks (prediction) using data simulated with the trained policy is a somewhat surprising and valuable outcome.

Concerns:
- The infraction set considered in this work is very minimalistic and only considers basic events: road departure and collisions. Traffic violations, in reality, are much more complex.
- Albeit FDE metric is a standard proxy for measuring realism, it is very indifferent toward the types of mistakes that are made by the traffic participants. In other words, different mistakes with the same metric might introduce drastically different outcomes for the behavior of surrounding agents. At this point, it would be interesting to see more sophisticated metrics.
- Although the results are quite interesting the main idea is not particularly groundbreaking and more or less incremental improvement upon the work "Imitation is not enough".

**Quality Of The Limitations Section:**

Limitations are addressed clearly

**Questions For Rebuttal:**

- The authors mentioned that the value network does not share parameters with the policy. Could the authors propose insights if this was detrimental to learning?
- The Fig. 3 is worth bringing to the unified scale across domains with a much finer grid step to see jumps in performance across different datasets.
- The authors mentioned early termination of episodes upon the infraction. Was it actually necessary? Shouldn't agents learn faster if allowed to continue simulation due to occasional recovery and, hence, receiving more detailed reward-based feedback?
- The baseline BC + RL seems to be the worst at imitation. It is a bit surprising. Could the authors provide a more detailed analysis of why this happens?
- How validation/testing is separated? It is especially interesting to see if testing was performed at the level of different logical scenarios vs just specific instances of the same logical scenarios.
- Do the authors plan to release the source code?
- Could the authors propose how their approach could be extended toward generating long-tail events automatically?
- Some citations are definitely missing:
  - PredictionNet: Real-Time Joint Probabilistic Traffic Prediction for Planning, Control, and Simulation. This work has quite extensive validation, including real driving, and combines RL and IL objectives to retain realism and reduce infractions.
  - BITS: Bi-level Imitation for Traffic Simulation. (I generally couldn't find any citations from M. Pavone group, although they have a lot of work toward traffic realism and prediction). This work also provides quite extensive ablation and goes beyond standard metrics for realism evaluation.

**Robotics Focus:**

Highly relevant to robotics but no hardware experiments

**Summary Of Paper:**

The paper proposes a method of combining RL and Imitation Learning objectives to improve the realism of simulated traffic agents in application to AV. To this end, the authors combine pre-collected datasets from human driving together with scripted long-tail scenarios for closed-loop learning with RL objective. To overcome problems of credit assignment in RL objective, the paper introduces agent-factorized rewards, policy, and value estimation (together with the advantage estimator).
The method is evaluated in simulated and held-out pre-recorded human driving, showing improvements in collision avoidance and reduction in FDE metric measuring realism of traffic.

**Summary Of Recommendation:**

The paper is enjoyable to read and introduces interesting contributions, tackles important problems, and considers a quite wide range of baselines for comparison. The overall results are quite valuable for RL practitioners in AV.
On the downside, it has a somewhat minimalistic scope and does not bring completely novel/ground-breaking ideas that could have a major impact on self-driving. Neither does it tackle the most intricate aspect of AV - long tail scenario generation.

Considering all that, I am inclined toward accepting the paper, but can't suggest a "strong accept" at this point.

---

> ### Author Response · Authors · 2023-08-10
> **Response to Reviewer Gvqu (1/3)**
>
> We thank the reviewer for the valuable comments, and respond below.
>
> **W1:**
> > The infraction set considered in this work is very minimalistic and only considers basic events: road departure and collisions. Traffic violations, in reality, are much more complex.
>
> Encoding complex traffic violations mathematically can be a difficult task [1] and remains an open problem. Road departures and collisions are perhaps some of the most severe violations and also the clearest to model mathematically. Hence in this work, we take the approach of encoding simple and obvious traffic infractions as sparse rewards and instead rely on imitation learning to learn the more complex subtleties of driving and avoid the more nuanced traffic violations. While shaping the reward with more complex traffic violations could improve performance, we believe it to be out of scope for this paper.
>
> **W2:**
> > Albeit FDE metric is a standard proxy for measuring realism, it is very indifferent toward the types of mistakes that are made by the traffic participants. In other words, different mistakes with the same metric might introduce drastically different outcomes for the behavior of surrounding agents. At this point, it would be interesting to see more sophisticated metrics.
>
> We complement the FDE metric with both infraction rates and JSD. Infractions like collisions and off-road are discrete events that can drastically change a scenario while having little impact on FDE. We also use JSD metrics which capture the distributional similarity between simulations and real data, which can capture changes like mode collapse, etc. Thus we believe our suite of metrics are among the most comprehensive in existing work and provide reasonable insight into our method and baselines.
>
> **W3:**
> > W3: Although the results are quite interesting the main idea is not particularly groundbreaking and more or less incremental improvement upon the work "Imitation is not enough".
>
> We make two novel contributions that allow us to significantly outperform the approach proposed in Imitation is not Enough [2].  We first note that [2] takes a simple approach towards combining IL and RL by using an open-loop BC+ closed-loop RL approach. In fact it is common in the literature to combine IL and RL using an open-loop IL loss and closed-loop RL, e.g. [5,7] . Unfortunately this leaves the policy susceptible to distribution shift with respect to imitation. We propose a principled approach towards performing IL+RL in closed-loop jointly, and our experiments show that this significantly outperforms the BC+RL approach taken in [2]. We provide more analysis as requested in our response to Q4.
>
> **Q1:**
> > The authors mentioned that the value network does not share parameters with the policy. Could the authors propose insights if this was detrimental to learning?
>
> In early experiments, we found not sharing parameters resulted in more stable training, likely due to preventing updates to the policy from interfering with the value function, and vice versa. We will add this point into the camera ready.
>
> **Q2:**
> > The Fig. 3 is worth bringing to the unified scale across domains with a much finer grid step to see jumps in performance across different datasets.
>
> Could the reviewer clarify that “unified scale across domains” means having the y axis consistent between the 4 plots in Figure 3, and “finer grid step” means more grid lines in the plot? If so, we thank the reviewer for the suggestion and will include this in the camera ready supplementary.
>
> **Q3:**
> > The authors mentioned early termination of episodes upon the infraction. Was it actually necessary? Shouldn't agents learn faster if allowed to continue simulation due to occasional recovery and, hence, receiving more detailed reward-based feedback?
>
> We treated infractions as terminal states in the MDP for a few reasons. Firstly for collision, it is unclear what the optimal behavior (or recovery) looks like after a collision. In the real world this usually involves stopping, etc., so we believe modeling it as an early termination is the correct approach. Similarly for driving off-road, the actor is likely in a state that it is physically impossible to recover from the real world, as an off-road event would imply the actor has driven off the shoulder into a divider. Finally, in early experiments, we found that continuing simulation for off-road events (and not modeling any shoulders or dividers, physics of off-road driving, etc.) would slow down training since in early phases the policy would drive offroad very early and very severely with no hope of recovering. Resetting in this case prevents wasted simulation in very out-of-distribution states where the policy is completely off the map, etc. We will include this discussion in the camera ready.

---

> ### Author Response · Authors · 2023-08-10
> **Response to Reviewer Gvqu (2/3)**
>
> **Q4:**
> > The baseline BC + RL seems to be the worst at imitation. It is a bit surprising. Could the authors provide a more detailed analysis of why this happens?
>
> The reason why BC+RL performs very poorly in imitation is because of the mixed open-loop (BC) and closed-loop (RL) objective. Because BC is open-loop, it only provides imitation regularization in expert states. In non-expert states, only the RL signal is present, and there is no imitation objective. As a result, the policy still suffers from distribution shift and compounding error with respect to the imitation objective. As you unroll the policy, small errors accumulate and push the policy into non-expert states. Here, the policy has never received IL regularization, leading to worse imitation. In contrast, our formulation gives rise to a joint closed-loop IL + RL that learns a policy more robust to compounding error, with respect to both imitation and infractions. We thank the reviewer for their feedback that this was not communicated well in the brief explanation in lines 236-238, and will expand upon it in the camera ready.
>
> **Q5:**
> > How validation/testing is separated? It is especially interesting to see if testing was performed at the level of different logical scenarios vs just specific instances of the same logical scenarios.
>
> We agree with the reviewer that this is an interesting point - the short answer is that we have done testing in both settings. To expand upon the brief description on L208-209, the results in Figure 3 split training/testing on the concrete scenario level, meaning that all scenarios in the test set use unique held out parameters that are i.i.d as the training parameters. This is akin to the standard supervised learning setup of measuring in-distribution generalization. We additionally present results in L244-249 which evaluate on held out logical scenarios, which contain very different scenarios (a car coming to a complete stop, etc.). The goal here is to measure whether the policy can further generalize to out-of-distribution scenarios. We will update the camera ready to be more clear in this regard.
>
> **Q6:**
> > Do the authors plan to release the source code?
>
> As the experiments are conducted using an in-house dataset and proprietary simulator, releasing the full source code is an extremely involved process and we currently do not have plans to do so. We will add an algorithm box and more detailed training hyperparameters to the supplementary to aid any reproduction efforts.
>
> **Q7:**
> > Could the authors propose how their approach could be extended toward generating long-tail events automatically?
>
> While this work uses a knowledge-based scenario generation approach (designing logical scenarios, automatically generating concrete scenarios), existing fully automated approaches can technically be used. For example, replacing our procedurally generated scenarios with scenarios found with adversarial methods [3,4] can be an approach. However, it is an open problem on how to generate scenarios that are actually useful for training. [3] take the approach of considering both realistic and solvable scenarios and show promising results, but additional considerations (coverage/diversity, difficulty, etc) could still be considered. We leave this line of inquiry for interesting future work.

---

> ### Author Response · Authors · 2023-08-10
> **Response to Reviewer Gvqu (3/3)**
>
> **Q8:**
> > Some citations are definitely missing
>
> We thank the reviewer for these references and will include discussions in the revised related work section. As the reviewer mentioned, [5] also combines RL and IL to retain realism while reducing infractions. In particular, an open-loop IL objective is used to train the main prediction module, while a model based RL objective is used to train an ego-motion planning header in synthetic cut-in scenarios. This approach results in impressive results, but contains some shortcomings which RTR can address. Firstly, similar to [2] and the discussion in Q4, the open-loop IL objective leaves the model susceptible to distribution shift, and techniques like the proposed drift correction technique are used to mitigate this. RTR takes a principled approach by learning completely in closed-loop. Secondly, the RL component and cut-in scenario is only used to train the ego motion head. In our experiments, we perform multiagent learning on all traffic agents in this fashion with a larger variety of procedurally generated long-tail scenarios and find that this improves realism for all agents, rather than only focusing on the ego agent.
>
> We also thank the reviewer for pointing us to [6] and the more extensive realism metrics they use. Our JSD metrics are similar to the Wasserstein distance metrics used in [6], which help capture how well simulations match the real world in distribution from a coverage/diversity perspective.
>
> **References:**
> [1] Shalev-Shwartz, Shai, Shaked Shammah, and Amnon Shashua. "On a formal model of safe and scalable self-driving cars." arXiv preprint arXiv:1708.06374 (2017).
> [2] Lu, Yiren, et al. "Imitation is not enough: Robustifying imitation with reinforcement learning for challenging driving scenarios." arXiv preprint arXiv:2212.11419 (2022).
> [3] Rempe, Davis, et al. "Generating useful accident-prone driving scenarios via a learned traffic prior." Proceedings of the IEEE/CVF Conference on Computer Vision and Pattern Recognition. 2022.
> [4] Hanselmann, Niklas, et al. "King: Generating safety-critical driving scenarios for robust imitation via kinematics gradients." European Conference on Computer Vision. Cham: Springer Nature Switzerland, 2022.
> [5] Kamenev, Alexey, et al. "Predictionnet: Real-time joint probabilistic traffic prediction for planning, control, and simulation." 2022 International Conference on Robotics and Automation (ICRA). IEEE, 2022.
> [6] Xu, Danfei, et al. "BITS: Bi-level imitation for traffic simulation." 2023 IEEE International Conference on Robotics and Automation (ICRA). IEEE, 2023.
> [7] TrajGEN: Generating realistic and diverse trajectories with reactive and feasible agent behaviors for autonomous driving IEEE Transactions on Intelligent Transportation Systems, 2022

---

> ### Author Response · Authors · 2023-08-15
> **Follow up**
>
> Dear reviewer, as the discussion phase is ending soon, we'd like to make sure if you have any additional concerns that we could discuss or address. Thank you!

---

### Decision · Program_Chairs · 2023-08-30

**Decision:**

Accept (Poster)

**Comment:**

The authors' response has addressed most of the reviewers' concerns, and the general assessment is that this paper provides a good contribution to CoRL. In the final version of this paper, the authors should carefully revise the paper to address all reviewers' comments / suggestions.